# REMOVING BIASES FROM MOLECULAR REPRESENTATIONS VIA INFORMATION MAXIMIZATION

**Chenyu Wang**[*1,2], **Sharut Gupta**[1], **Caroline Uhler**[2,3], **Tommi Jaakkola**[1]

[1]Computer Science and Artificial Intelligence Laboratory, Massachusetts Institute of Technology
[2]Eric and Wendy Schmidt Center, Broad Institute of MIT and Harvard
[3]Laboratory for Information and Decision Systems, Massachusetts Institute of Technology

## ABSTRACT

High-throughput drug screening – using cell imaging or gene expression measurements as readouts of drug effect – is a critical tool in biotechnology to assess and understand the relationship between the chemical structure and biological activity of a drug. Since large-scale screens have to be divided into multiple experiments, a key difficulty is dealing with batch effects, which can introduce systematic errors and non-biological associations in the data. We propose InfoCORE, an **Info**rmation maximization approach for **CO**nfounder **RE**moval, to effectively deal with batch effects and obtain refined molecular representations. InfoCORE establishes a variational lower bound on the conditional mutual information of the latent representations given a batch identifier. It adaptively reweighs samples to equalize their implied batch distribution. Extensive experiments on drug screening data reveal InfoCORE's superior performance in a multitude of tasks including molecular property prediction and molecule-phenotype retrieval. Additionally, we show results for how InfoCORE offers a versatile framework and resolves general distribution shifts and issues of data fairness by minimizing correlation with spurious features or removing sensitive attributes. The code is available at https://github.com/uhlerlab/InfoCORE.

## 1 INTRODUCTION

Representation learning (Bengio et al., 2013) has become pivotal in drug discovery (Wu et al., 2018) and understanding biological systems (Yang et al., 2021b). It serves as a pillar for recognizing drug mechanisms, predicting a drug's activity and toxicity, and identifying disease-associated chemical structures. A central challenge in this context is to accurately capture the nuanced relationship between the chemical structure of a small molecule and its biological or physical attributes. Most molecular representation learning methods only encode a molecule's chemical identity and hence provide unimodal representations (Wang et al., 2022; Xu et al., 2021b). A limitation of such techniques is that molecules with similar structures can have very different effects in the cellular context.

High-content drug screens have been developed that output post-perturbation (i.e., after the application of a drug) cellular images and gene expression (Chandrasekaran et al., 2021). Such datasets provide a unique opportunity to improve our understanding of the biological effect of a compound and help refine the representation of molecules. Given the huge chemical space of over $10^{60}$ molecules of possible interest for drug discovery (Reymond et al., 2010), the full space cannot be explored experimentally. Thus, computational methods are needed to obtain biologically informed molecular representations that can be generalized to untested molecules.

Recent works in this direction (Nguyen et al., 2023; Zheng et al., 2022) focused on training models to map 2D molecular structures to high-content cell microscopy images (Bray et al., 2017; Chandrasekaran et al., 2023) through multimodal contrastive learning (Radford et al., 2021). Generalizing across molecular structures has been challenging due to the variability in the training data as drug responses vary under different conditions. In particular, batch effects are pervasive, stemming from the fact that large-scale drug screens have to be divided into multiple experiments over many years.

---

[*]correspondence to wangchy@mit.edu

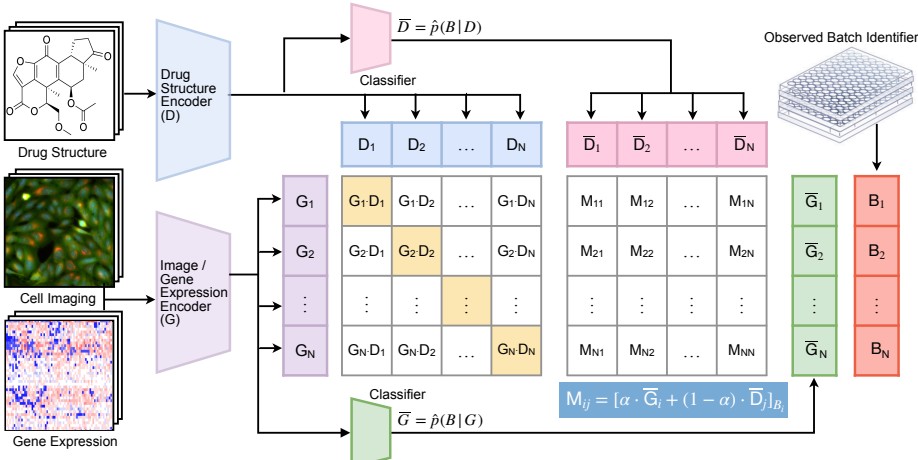

Figure 1: The model takes drug structure and screening data (gene expression or cellular imaging) as input and encodes it into latent representations $D_i$ and $G_i$. Two batch classifiers are trained to obtain the posterior batch distribution $\bar{G}_i = \hat{p}(B|G_i)$ and $\bar{D}_i = \hat{p}(B|D_i)$. The two encoders are jointly trained to predict the correct pairings of molecule-phenotype training samples. The loss $L$ is a function of two matrices: the pairwise cosine similarity of the representations (left) and the weight matrix for each pair, where the weight $M_{ij}$ is the weighted (with scaler $\alpha$) average posterior probability of the latent representations being in batch $B_i$, i.e. $\bar{G}_i[B_i]$ and $\bar{D}_j[B_i]$ (right). Sample pairs with more similar posterior batch distributions are emphasized more in $L$. The figure illustrates the term in the loss $L$ using $G$ as an anchor; using $D$ as an anchor can be done analogously.

Batch effects refer to non-biological associations introduced through the measurement process. They can systematically distort the data and make it challenging to isolate the true biological signal.

While various statistical approaches have been proposed to correct for batch effects (Johnson et al., 2007; Korsunsky et al., 2019), these generally consist of a preprocessing step that is independent of the downstream training task. We provide a more effective batch correction method by directly integrating it with the learning algorithm, framing the task similar to removing the impact from a sensitive attribute (batch number). While various techniques have been developed for this problem, including in computer vision, they do not directly address the challenges in drug screening. For example, when observations of batch-drug combinations are limited in number and coverage, using contrastive samples with the same value of the sensitive attribute as in Ma et al. (2021) can lead to poor generalization. Zhang et al. (2022) rely on models to generate unbiased image training data, but generative models for high-content screening data are in their infancy (Yang et al., 2021a).

In this paper, we introduce a novel method, InfoCORE, designed to mitigate confounding factors in multimodal contrastive learning. While we describe our method in the context of batch effect removal for molecular representation learning, we also show its effectiveness as a general-purpose framework, such as for the removal of sensitive information to enhance fairness. InfoCORE formulates an intuitive and easy-to-optimize objective based on a variational lower bound on conditional mutual information. In essence, sample pairs with more similar batch distributions are emphasized more in the InfoNCE loss function (Oord et al., 2018). This weighting scheme enables the model to adapt its learning strategy for each sample based on the batch-related information present in the latent representations. The proposed framework is illustrated in Figure 1.

We conduct experiments with two common readouts of high-content drug screens: LINCS gene expression profiles (Subramanian et al., 2017) and cell imaging profiles (Bray et al., 2017). We show that InfoCORE consistently outperforms the baseline models across a range of downstream tasks, including molecule-phenotype retrieval and molecular property prediction. Furthermore, our empirical evaluations demonstrate InfoCORE's broad applicability well beyond drug screening. In particular, we demonstrate that InfoCORE obtains improved representations with respect to various fairness measures over existing baselines across multiple datasets, including UCI Adult (Asuncion & Newman, 2007), Law School (Wightman, 1998), and Compas (Angwin et al., 2022).

To summarize, the main contributions of our work are:

- We propose InfoCORE, a framework for multimodal molecular representation learning, capable of integrating diverse high-content drug screens with chemical structures.
- Theoretically, we show that InfoCORE maximizes the variational lower bound on the conditional mutual information of the representation given the batch identifier. It empirically outperforms various baselines on tasks such as molecular property prediction and molecule-phenotype retrieval.
- The information maximization principle of InfoCORE extends beyond drug discovery. We empirically demonstrate its efficacy in eliminating sensitive information for representation fairness.

## 2 METHOD

In this section, we introduce InfoCORE, provide the intuition for how it counteracts biases from irrelevant attributes, and derive the corresponding training objective. In Section 2.1, we describe the underlying graphical model as well as the main learning objective, which is based on conditional mutual information. In Section 2.2, we establish a tractable lower bound for this objective, which we show can be interpreted as the InfoNCE loss (Oord et al., 2018), but with unequally weighted negative samples. See Appendix E for a short review and additional backgrounds on InfoNCE.

### 2.1 CONDITIONAL MUTUAL INFORMATION MAXIMIZATION TO REMOVE BIASES FROM IRRELEVANT ATTRIBUTES.

Figure 2 depicts the graphical model for both the observations and the learned representations. In this model, $(X_d, X_g, X_b)$ represent observed variables – $X_d$ signifies the molecular structure of a drug, $X_g$ indicates the shift in gene expression or other phenotype induced by applying the drug, and $X_b$ represents an irrelevant attribute such as the experimental batch in the context of molecular representation learning. Illustrated by dashed lines in Figure 2, $(Z_d, Z_g)$ are representations generated from the trained encoders: $Z_d = \text{Enc}_d(X_d; \theta_d)$, $Z_g = \text{Enc}_g(X_g; \theta_g)$, where $\text{Enc}_d(.; \theta_d)$ is the encoder of the drug structure and $\text{Enc}_g(.; \theta_g)$ is the encoder for the screening readout.

The graphical model illustrates that $X_b$ is a confounding factor affecting the learned representations. The observed phenotype $X_g$ is influenced by the drug's effect, derived from its molecular structure, but also by external non-biological factors (i.e. batch effects) $X_b$. Since batch assignment in biological experiments is not always (often) fully randomized, the batch number can affect which drug is applied, explaining the arrow from $X_b$ to $X_d$. The presence of a backdoor path from $X_d$ to $X_g$ through $X_b$ means that naive estimates of the effect of $X_d$ on $X_g$ will be biased.

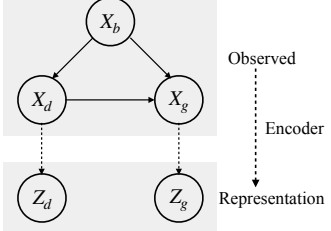

Figure 2: Graphical Model

We adopt a conditional mutual information objective to address this challenge, conditioned on the irrelevant attributes $X_b$. This results in learned *latent* representations $Z_d, Z_g$, where variations associated with $X_b$ are effectively excluded. Intuitively, this is because the features tied to $X_b$ do not improve the conditional mutual information objective, and because the latent representation has a finite dimension (implying a representational resource constraint). This claim is further justified in Ma et al. (2021) and Proposition 2 of Robinson et al. (2021). More precisely, based on the graphical model given above, we specify a conditional mutual information criterion for learning latent representations of $X_d$ and $X_g$ that do not contain the effect of $X_b$. This criterion satisfies the following bound[1]:

$$\max_{\theta_d, \theta_g} \frac{1}{2} \left( I(Z_d; X_g | X_b; \theta_d) + I(Z_g; X_d | X_b; \theta_g) \right) \geq \max_{\theta_d, \theta_g} I(Z_d; Z_g | X_b; \theta_d, \theta_g). \quad (1)$$

As noted earlier, this objective function emphasizes drug's bioactivity by focusing on shared features of the two modalities that are unrelated to batch.

### 2.2 REWEIGHTED INFONCE OBJECTIVE AS TRACTABLE LOWER BOUND.

Direct estimation of conditional mutual information is computationally intractable for high-dimensional continuous random variables. The InfoNCE objective in contrastive learning (Poole

---

[1]The inequality is based on the Markov relationship in the graphical model and data processing inequality.

et al., 2019; Oord et al., 2018; Tian et al., 2020) has been introduced as a tractable alternative and was extended to the multimodal case in CLIP (Radford et al., 2021). It optimizes the representation to bring each "anchor" data point close to its paired "positive" and away from "negative" samples in the latent space (details in Appendix E). It was further extended to the conditional case by Ma et al. (2021), where the relationship between conditional mutual information and conditional contrastive learning (CCL) was built. In CCL, the negative samples are drawn from the conditional marginal distributions of the representations $z_d$ and $z_g$, $p(z_d|x_b)$ and $p(z_g|x_b)$, conditioned on the anchor's batch number $x_b$. The downside is that when there are limited observations for a particular value of $x_b$ (limited number of drugs), the small number of possible negative samples will lead to poor generalization to new molecular structures.

Table 1: Comparison of various methods such as CCL and CLIP with InfoCORE

| Method | CLIP | CCL | Ours |
|---|---|---|---|
| Debiasing | ✗ | ✓ | ✓ |
| Large Supply of Negatives | ✓ | ✗ | ✓ |

To address this challenge, we introduce InfoCORE, which reweighs the negative samples in the InfoNCE objective differentially based on the posterior batch distribution. This acts as a practical lower bound for the conditional mutual information $I(Z_d; Z_g|X_b)$ and dynamically adjusts the weight for each anchor-negative pair. A comparison of various methods is provided in Table 1.

### 2.2.1 SINGLE-SAMPLE LOWER BOUND OF CONDITIONAL MUTUAL INFORMATION.

InfoNCE (Oord et al., 2018) is one of the most popular objectives in contrastive learning. Poole et al. (2019) showed that the InfoNCE objective is a lower bound on mutual information. We obtain the bound in Proposition 1 by extending the variational lower bound from the energy-based variational family in Poole et al. (2019) to conditional mutual information. The proof is given in Appendix A.

**Proposition 1.** *Given random variables $Z_g, Z_d$ as representations of two data modalities and $X_b$ as the irrelevant attribute, the conditional mutual information $I(Z_d; Z_g|X_b)$ between representations conditioned on the irrelevant attribute has the following lower bound:*

$$I(Z_d; Z_g|X_b) \geq \mathbb{E}_{p(z_d, z_g, x_b)}[h(z_d, z_g, x_b)] - e^{-1}\mathbb{E}_{p(z_d)}\mathbb{E}_{p(z_g, x_b)}\left[e^{h(z_d, z_g, x_b)}\right] - I(Z_d; X_b), \quad (2)$$

*which results from using an energy-based variational family $q(z_g, x_b|z_d)$ to approximate the distribution $p(z_g, x_b|z_d)$:*

$$q(z_g, x_b|z_d) = \frac{p(z_g, x_b)}{Z(z_d)}e^{h(z_d, z_g, x_b)}, \text{ where } Z(z_d) = \mathbb{E}_{p(z_g, x_b)}\left[e^{h(z_d, z_g, x_b)}\right] \text{ is the partition function,}$$

*with equality holding when the critic[2] $h$ satisfies $h^*(z_d, z_g, x_b) = 1 + \log \frac{p(z_g, x_b|z_d)}{p(z_g, x_b)}$.*

This bound provides a tractable estimator – where the first two terms can be estimated given samples from the joint distribution and product marginal distribution, and $I(Z_d; X_b)$ can be optimized via adversarial training as shown by Guo et al. (2023). However, as explained in Poole et al. (2019), this bound may exhibit high variance due to its reliance on the upper bounds of the log partition function, i.e., $\log Z(z_d) \leq e^{-1}Z(z_d)$, which introduces high variance in its sample approximations.

### 2.2.2 INFOCORE AS A MULTI-SAMPLE LOWER BOUND OF CONDITIONAL MUTUAL INFORMATION.

To reduce the variance of the single sample lower bound discussed above, we develop a method that makes use of additional samples from the data. An intuitive approach is to extend the unconditional mutual information proposed in Oord et al. (2018) and Poole et al. (2019). However, applying this strategy directly in our use case would mean an adversarial optimization of $I(Z_d; X_b)$ (see Equation 2), which is disadvantageous from a computational perspective. Instead, we propose to decompose the distribution $p(z_g, x_b|z_d)$ into the product of a conditional distribution $p(z_g|z_d)$ – which is independent of batch $x_b$ – and the posterior batch distribution based on the latent representation $p(x_b|z_d, z_g)$. We show that by separately approximating these distributions, maximizing the conditional mutual information maintains connections to the widely-used InfoNCE objective.

In the following propositions, we build on the decomposition of $p(z_g, x_b|z_d)$ to derive a multi-sample lower bound of the conditional mutual information objective shown in Equation 3. The proof is given

---

[2]We follow the terminology in Poole et al. (2019) to refer to $h(z_g, z_g, x_b)$ as critic.

in Appendix B. Variables with superscript 1 (i.e. $Z_d^1, Z_g^1, X_b^1$) denote the corresponding variables for the anchor-positive pair. Those with superscript other than 1 (i.e. $Z_d^{2:K}, Z_g^{2:K}$) denote the negative samples in the multi-sample case; $K$ is the number of negative samples. A lookup table of the definition of all variables is provided in Appendix J.

**Proposition 2.** *Given samples $(z_d^1, z_g^1, x_b^1)$ drawn from the joint distribution $(Z_d^1, Z_g^1, X_b^1) \sim p(z_d, z_g, x_b)$ and $z_d^{2:K}$ drawn i.i.d. from the marginal distribution $Z_d^i \sim p(z_d)$ for $i = 2, ..., K$, then the conditional mutual information $I(Z_d^1; Z_g^1 | X_b^1)$ has the following lower bound:*

$$I(Z_d^1; Z_g^1 | X_b^1) \geq -L_{CLIP} - L_{CLF} + C - H(X_b^1), \quad (3)$$

*where* $L_{CLIP} = -\frac{1}{2} \left[ \mathbb{E}_{p(z_d^1, z_g^1, x_b^1) p(z_d^{2:K})} \left[ \log \frac{e^{f(z_g^1, z_d^1)}}{\frac{1}{K} \sum_{i=1}^{K} e^{f(z_g^1, z_d^i)} \cdot \hat{p}_g(x_b^1 | z_g^1, z_d^i)} \right] \right.$

$\left. + \mathbb{E}_{p(z_d^1, z_g^1, x_b^1) p(z_g^{2:K})} \left[ \log \frac{e^{f(z_g^1, z_d^1)}}{\frac{1}{K} \sum_{i=1}^{K} e^{f(z_g^i, z_d^1)} \cdot \hat{p}_d(x_b^1 | z_g^i, z_d^1)} \right] \right],$

$L_{CLF} = \frac{1}{2} \left[ \mathbb{E}_{p(z_d^1)} \left[ D_{\mathrm{KL}} \left( p \left( x_b^1 | z_d^1 \right) \| \hat{p} \left( x_b^1 | z_d^1 \right) \right) \right] + \mathbb{E}_{p(z_g^1)} \left[ D_{\mathrm{KL}} \left( p \left( x_b^1 | z_g^1 \right) \| \hat{p} \left( x_b^1 | z_g^1 \right) \right) \right] \right],$

$C = \frac{1}{2} \mathbb{E}_{p(z_d^1, z_g^1, x_b^1)} \left[ \log \frac{\hat{p}_g(x_b^1 | z_g^1, z_d^1) \cdot \hat{p}_d(x_b^1 | z_g^1, z_d^1)}{\hat{p}(x_b^1 | z_g^1) \cdot \hat{p}(x_b^1 | z_d^1)} \right].$

*This holds for any choice of critic $f(z_g, z_d)$ and variational distribution $\hat{p}_g(x_b | z_g, z_d)$, $\hat{p}_d(x_b | z_g, z_d)$, with equality holding when $f^*(z_d, z_g) = \log \frac{p(z_g | z_d)}{p(z_g)}$ and $\hat{p}_g^*(x_b | z_g, z_d) = \hat{p}_d^*(x_b | z_g, z_d) = p(x_b | z_g, z_d)$.*[3]

We note that $L_{\mathrm{CLIP}}$ resembles the symmetrical InfoNCE loss in multimodal contrastive learning (Radford et al., 2021), but with the negative samples reweighted according to the posterior batch distributions; i.e., the negative samples $z_d^i$ that are more likely to be in the anchor's batch $x_b^1$ (having higher value $\hat{p}_g(x_b^1 | z_g^1, z_d^i)$) are weighted more. $L_{\mathrm{CLF}}$ denotes the average classification loss achieved when training a classifier to predict the batch number from a given latent representation ($z_g$ or $z_d$).

**Posterior Batch Distribution Estimation.** Estimating the lower bound in Proposition 2 requires estimating the reweighting factors $\hat{p}_g(x_b^1 | z_g^1, z_d^i)$ and $\hat{p}_d(x_b^1 | z_d^1, z_g^i)$. This presents several challenges, especially for negative samples when $i \neq 1$: 1) The corresponding empirical observations are absent, leading to a lack of data; 2) Since most inputs $z_d$ and $z_g$ are unpaired, directly training a classifier with the paired input `concat`$[z_g^i, z_d^1]$ could result in poor out-of-distribution generalization. Additionally, this approach would require the computationally intensive step of rerunning the classifier for each pair separately.

Given that $p(x_b^1 | z_g^1, z_d^i)$ represents the posterior batch distribution informed by both latent representations, both modalities offer valuable but possibly overlapping information about the batch number $x_b^1$. Hence, we can estimate $p(x_b^1 | z_g^1, z_d^i)$ using the weighted arithmetic average of the posteriors given each individual latent, $\hat{p}(x_b^1 | z_g^1)$ and $\hat{p}(x_b^1 | z_d^i)$, which measures the remaining batch-related information in each latent representation. This offers an intuitive and computationally inexpensive estimate for $p(x_b^1 | z_g^1, z_d^i)$:

$$\hat{p}_g(x_b^1 | z_g^1, z_d^i) = \alpha \cdot \hat{p}(x_b^1 | z_g^1) + (1 - \alpha) \cdot \hat{p}(x_b^1 | z_d^i), \; \hat{p}_d(x_b^1 | z_g^i, z_d^1) = \alpha \cdot \hat{p}(x_b^1 | z_d^1) + (1 - \alpha) \cdot \hat{p}(x_b^1 | z_g^i).$$

Based on this, the denominator in $L_{\mathrm{CLIP}}$ can be viewed as a combination of uniformly weighted negative samples $\alpha \cdot \hat{p}(x_b^1 | z_g^1) \cdot \frac{1}{K} \sum_{i=1}^{K} e^{f(z_g^1, z_d^i)}$ and $\alpha \cdot \hat{p}(x_b^1 | z_d^1) \cdot \frac{1}{K} \sum_{i=1}^{K} e^{f(z_d^1, z_g^i)}$, as well as confounder biased weighted negative samples $(1 - \alpha) \frac{1}{K} \sum_{i=1}^{K} \hat{p}(x_b^1 | z_d^i) \cdot e^{f(z_g^1, z_d^i)}$ and $(1 - \alpha) \frac{1}{K} \sum_{i=1}^{K} \hat{p}(x_b^1 | z_g^i) \cdot e^{f(z_d^1, z_g^i)}$. $\alpha$ serves as a tuning parameter, determining the degree of reliance on the posterior estimated from the common anchor ($z_d^1$ or $z_g^1$). It presents a tradeoff between correcting for irrelevant attributes and enhancing model generalization by incorporating a larger supply of negative samples. When $\alpha = 1$, $L_{\mathrm{CLIP}}$ collapses to the unweighted InfoNCE loss in CLIP (Radford et al., 2021). The following proposition shows that under these assumptions, $C$ can be lower bounded by 0, thus simplifying the lower bound further. The proof is given in Appendix C.

**Proposition 3.** *When estimating $\hat{p}_g(x_b^1 | z_g^1, z_d^i)$ as the weighted average of $\hat{p}(x_b^1 | z_g^1)$ and $\hat{p}(x_b^1 | z_d^i)$, and analogously for $\hat{p}_d(x_b^1 | z_g^i, z_d^1)$, the term $C$ defined in Proposition 2 is lower bounded by zero.*

---

[3] We note that $\hat{p}(x_b | z_g)$ and $\hat{p}(x_b | z_d)$ cancel out in $-L_{\mathrm{CLF}} + C$, but we spell out these terms for ease of estimating $C$ later.

Based on Proposition 3, the lower bound in Proposition 2 can be further reduced to $I(Z_d^1; Z_g^1 | X_b^1) \geq -L_{\text{CLIP}} - L_{\text{CLF}} - H(X_b^1)$. Since $H(X_b^1)$ is a constant, maximization of the conditional mutual information lower bound gives rise to the minimization of our InfoCORE loss function:

$$L_{\text{InfoCORE}} = L_{\text{CLIP}} + L_{\text{CLF}}, \tag{4}$$

where $L_{\text{CLIP}}$ and $L_{\text{CLF}}$ are defined as in Proposition 2.

Estimating the batch distribution as opposed to directly utilizing the observed values of $X_b$ for the negative samples as in Ma et al. (2021) and Tsai et al. (2021) has advantages: Since experimental batches may contain molecules that share scaffolds, while others may possess randomly assigned ones, the batch confounding effect can vary from batch to batch. InfoCORE can deal with the fact that positive and negative samples may be affected differently by the batch confounder. Moreover, since batch distribution is estimated using the latent representations and not the original data, once the batch effect has been mitigated, InfoCORE implicitly adjusts during training and ceases to reweight the negative samples.

### 2.2.3 COMPUTATIONAL CONSIDERATIONS.

We iteratively optimize the loss function by updating the encoders based on $L_{\text{CLIP}}$ and then the classifiers based on $L_{\text{CLF}}$. Note that although we freeze the classifiers when optimizing $L_{\text{CLIP}}$, $\hat{p}(x_b^1 | z_d^i)$ and $\hat{p}(x_b^1 | z_g^i)$ depend on the encoder parameters, which introduces competing objectives. Precisely, the gradient of $L_{\text{CLIP}}$ w.r.t. the representations $z_d^i$ and $z_g^i$ can be decomposed into two components: a standard component as in CLIP (Radford et al., 2021), involving $\partial f(z_d^1, z_g^i)/\partial z_g^i$ and $\partial f(z_g^1, z_d^i)/\partial z_d^i$, and a competing component involving $\partial \hat{p}(x_b^1 | z_g^i)/\partial z_g^i$, and $\partial \hat{p}(x_b^1 | z_d^i)/\partial z_d^i$ (see details in Appendix D). Note that both gradient components drive the representations towards reduced batch effect: In the standard component, samples with similar batch distributions are up-weighted and thus the latent variables are driven to be less informative of batch number. In the other component, when $i = 1$, the derivatives $\partial \hat{p}(x_b^1 | z_g^1)/\partial z_g^1$ and $\partial \hat{p}(x_b^1 | z_d^1)/\partial z_d^1$ update the encoder to make $\hat{p}(x_b^1 | z_d^1)$ and $\hat{p}(x_b^1 | z_g^1)$ smaller, making the latent representations less informative of batch number. When $i > 1$, according to the confounder-biased weighted part of $L_{\text{CLIP}}$'s denominator, i.e. $(1 - \alpha)\frac{1}{K} \sum_{i=1}^K \hat{p}(x_b^1 | z_d^i) \cdot e^{f(z_g^1, z_d^i)}$, we can infer that $\hat{p}(x_b^1 | z_d^i)$ is adjusted similar in response to $e^{f(z_g^1, z_d^i)}$ as $e^{f(z_g^1, z_d^i)}$ is adjusted on the basis of $\hat{p}(x_b^1 | z_d^i)$. Therefore, the gradient updating schema of $\hat{p}(x_b^1 | z_d^i)$ in terms of the underlying representation $z_d^i$ is, in broad terms, consistent with that of $e^{f(z_g^1, z_d^i)}$. This motivates our approach to use either gradient or stop gradient on $\hat{p}$ and treat them as constant weights. In practice, we use a hyperparameter $\lambda$ to control the gradient update schedule.

## 3 EXPERIMENTS

In this section, we present comprehensive experiments demonstrating the effectiveness of InfoCORE to remove biases in representation learning. Across experiments, we compare InfoCORE with the unconditional multi-modal contrastive learning method CLIP (Radford et al., 2021; Nguyen et al., 2023) and the recent conditional contrastive learning method CCL (Ma et al., 2021).

### 3.1 SIMULATION STUDY

To offer insights into InfoCORE's capability to discern a drug's biological effect from noisy data influenced by batch confounders, we conduct a low-dimensional simulation experiment that replicates our data generation process. Here, each drug, associated with a real effect from one of five categories, is randomly assigned to one of 25 batches, with a total of 50 drugs per batch. The observational data $X_g$, corresponding to screen output, and $X_d$, corresponding to drug structure, are 10-dimensional vectors. These vectors are generated by a randomized function of the input which concatenates biological effect, batch effect, and Gaussian noise. The dataset is split, with half utilized for training and the remaining half held out for evaluation. Additional details pertaining to data generation and experiments can be found in Appendix H.

The 2-dimensional latent representation for the drug screens, $Z_g$, learned by different methods for the held-out data and several quantitative metrics (Xu et al., 2021a) are depicted in Figure 3; the corresponding figure for drug structure and a detailed explanation of these metrics are provided in

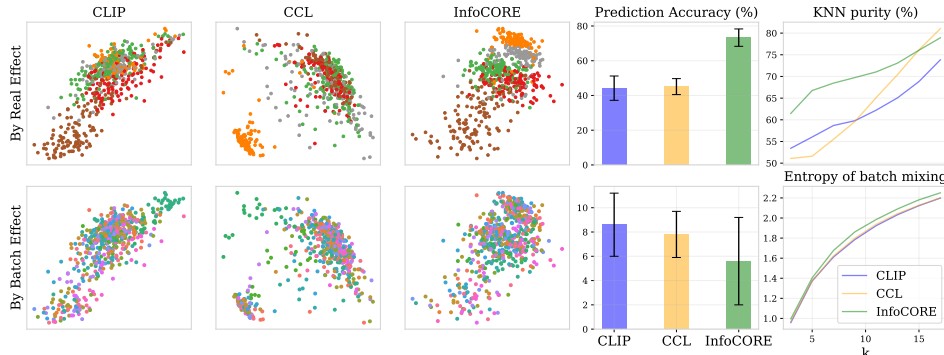

Figure 3: 2D representation visualizations and quantitative metrics for each method: top row uses real effect category for coloring and calculating prediction accuracy; bottom row uses batch identifier. The rightmost column shows KNN purity and entropy of batch mixing across different k values.

Appendix I. An ideal representation should distinctly separate real effect categories without being influenced by batch identifier. This is characterized by high accuracy in predicting real effects, low accuracy for batch identifiers, and elevated values of KNN purity and entropy of batch mixing. CLIP is affected by batch confounders and has difficulty identifying real effect categories, while CCL's embeddings, despite fusing batch identifiers, falter in distinguishing biological effects due to limited negative samples. Conversely, InfoCORE achieves a balance between debiasing and expressivity, offering superior representations that discern real effects and effectively mitigate batch effects.

## 3.2 Representation Learning of Small Molecules.

**Dataset Description.** We use two high-content drug screening datasets, L1000 gene expression profiles (Subramanian et al., 2017) (GE) and cell imaging profiles obtained from the Cell Painting assay (Bray et al., 2017) (CP). In GE, we select data from the nine core cell lines, resulting in 17,753 drugs and 82,914 drug-cell line pairs. In CP, 30,204 small molecules are screened in one cell line (U2OS). We use the hand-crafted image features obtained by the popular CellProfiler method (McQuin et al., 2018). The chemical structures are featurized using Mol2vec (Jaeger et al., 2018).

**Molecule-Phenotype Retrieval.** The drug repurposing or drug discovery task can be viewed as follows: identify molecules (e.g., from a drug repurposing library) that are most likely to induce a given desired phenotypic change (i.e., gene expression change from diseased towards normal). To mimic this process, similar to the setting in Zheng et al. (2022), we randomly split both datasets into a training set consisting of 80% of the molecules and hold out the remaining molecules for testing. We use top N accuracy (N=1, 5, 10) as an evaluation metric and analyze two retrieval libraries: (a) all molecules in the held-out set (*whole*), and (b) held-out set molecules that are in the same experimental batch as the retrieving target (*batch*). Accuracies over both libraries as a whole reflect the model's ability of molecule-phenotype retrieval for drug discovery and drug repurposing.

Table 2: Retrieving accuracy of different methods for gene expression and cell imaging screens.

| Dataset | Gene Expression (GE) | | | | | | Cell Painting (CP) | | | | | |
|---|---|---|---|---|---|---|---|---|---|---|---|---|
| Retrieval Library | *whole* | | | *batch* | | | *whole* | | | *batch* | | |
| Top N Acc (%) | N=1 | N=5 | N=10 | N=1 | N=5 | N=10 | N=1 | N=5 | N=10 | N=1 | N=5 | N=10 |
| Random | 0.03 | 0.13 | 0.27 | 1.58 | 7.90 | 15.81 | 0.02 | 0.08 | 0.17 | 1.59 | 7.97 | 15.94 |
| CLIP | 5.96 | 18.59 | 27.17 | 12.23 | 30.29 | 42.63 | **7.23** | **20.95** | **28.89** | 13.20 | 37.78 | 52.72 |
| CCL | 1.93 | 5.85 | 8.37 | 12.76 | 32.39 | 45.77 | 1.31 | 4.93 | 7.38 | 13.20 | 37.99 | 53.13 |
| InfoCORE | **6.39** | **18.99** | **27.18** | **14.03** | **33.63** | **46.78** | 6.93 | 20.65 | 28.22 | **13.26** | **38.50** | **53.13** |

Since CLIP and CCL are not specifically designed for cross-modal molecular representation learning, we made several modifications to the vanilla algorithms for this task; see details in Appendix F. As shown in Table 2, InfoCORE is the only model that has strong performance in both libraries, especially in GE, which agrees with the belief that gene expression data is more affected by batch

effects. Batch-related features dominate the CLIP representation, leading to the model's poor performance in *batch*. For example, InfoCORE outperforms CLIP in GE over *batch* with a 15% increase in top 1 accuracy and 11% in top 5 accuracy. CCL's reliance on the few hundred negative samples within the same experimental batches hinders its out-of-batch generalization, leading to its poor performance in *whole*. Standard deviations over 3 random seeds are provided in Appendix I.

**Transfer Learning for Property Prediction.** Our trained model can be fine-tuned for various downstream tasks. Given that InfoCORE is pre-trained using drug screening data that contains information on the biological effect of a drug, transfer learning is expected to do well on the prediction of bioactivity-related properties. We test this by analyzing the following classification and regression tasks: For classification, we select 7 bioactivity-related benchmarks from MoleculeNet (Wu et al., 2018) and follow the standard scaffold splitting procedure as suggested by Hu et al. (2020). Since the regression tasks in MoleculeNet (i.e. ESOL, Lipo, FreeSolv) are not directly bioactivity-related, we use post-perturbation cell viability in the PRISM dataset (Corsello et al., 2020) for the regression task using the same scaffold splitting procedure. Further details are provided in Appendix H.

We analyze InfoCORE when pretraining using the two different drug screening datasets and compare the performance to CLIP and CCL. We report the performance of Mol2vec (Jaeger et al., 2018) with randomly initialized MLP of the same neural network architecture as a baseline for no-pretraining. Mean AUC-ROC is reported for the classification tasks and $R^2$ for the regression tasks, together with standard deviations based on 3 random seeds. The results are shown in Table 3. Compared to Mol2vec, the results clearly demonstrate the benefit of pretraining using drug screening data: significant gains are obtained in all properties except SIDER. Moreover, among the different pretraining strategies, InfoCORE is highly competitive, achieving the best performance in almost all properties when using GE, especially in ClinTox, HIV, and BACE, and most properties with CP.

Table 3: Performance of different methods on molecular property prediction task.

| | | Classification (ROC-AUC %) ↑ | | | | | | | | Reg ($R^2$ %) ↑ |
|---|---|---|---|---|---|---|---|---|---|---|
| | Datasets | BBBP | BACE | ClinTox | Tox21 | ToxCast | SIDER | HIV | Avg. | PRISM |
| | # Molecules | 2039 | 1513 | 1478 | 7831 | 8575 | 1427 | 41127 | - | 3172 |
| | # Tasks | 1 | 1 | 2 | 12 | 617 | 27 | 1 | - | 5 |
| | Mol2vec | 70.7(0.4) | 82.9(0.7) | 84.9(0.3) | 76.0(0.1) | 74.4(0.5) | 64.9(0.3) | 77.7(0.1) | 75.9 | 8.5(0.7) |
| GE | CLIP | 73.5(0.4) | 86.1(0.4) | 89.6(2.1) | 77.3(0.0) | 75.7(0.6) | 63.7(0.6) | 77.7(0.6) | 77.6 | 13.9(0.4) |
| | CCL | 73.0(0.8) | 85.9(0.6) | 90.5(1.0) | 77.0(0.2) | **75.8(0.2)** | 63.4(0.5) | 77.5(0.9) | 77.6 | **16.0(0.5)** |
| | InfoCORE | **73.5(0.3)** | **86.6(0.3)** | **91.9(1.9)** | **77.4(0.4)** | 75.7(0.2) | **64.8(0.6)** | **78.5(0.2)** | **78.3** | 14.8(0.1) |
| CP | CLIP | 73.4(0.8) | **85.2(0.4)** | 87.3(0.1) | 76.4(0.1) | 76.7(0.1) | 64.8(0.6) | 78.2(0.4) | 77.4 | 16.2(0.2) |
| | CCL | 73.7(0.5) | 84.9(0.9) | 87.7(1.8) | 75.9(0.3) | 75.7(0.4) | 65.2(0.4) | **79.3(0.3)** | 77.5 | 14.7(0.3) |
| | InfoCORE | **74.0(0.8)** | 85.0(0.2) | **89.3(0.5)** | 76.6(0.1) | **76.9(0.1)** | 65.2(0.1) | 78.7(0.1) | **78.0** | **16.2(0.3)** |

### 3.3 REPRESENTATION FAIRNESS.

**Dataset and Fairness Criteria.** In the following, we demonstrate InfoCORE's broad applicability by conducting experiments in representation fairness using three fairness datasets: UCI Adult (Asun-

Table 4: Performance of various methods on representation fairness task.

| | UCI Adult | | | Law School | | | Compas | | |
|---|---|---|---|---|---|---|---|---|---|
| Method | Acc↑ | EO↓ | EOPP↓ | Acc↑ | EO↓ | EOPP↓ | Acc↑ | EO↓ | EOPP↓ |
| CLIP | 85.1(0.1) | 20.7(1.8) | 15.2(1.7) | **83.1(0.2)** | 30.9(1.4) | 7.9(0.8) | **60.8(2.3)** | 18.4(2.4) | 11.7(1.9) |
| CCL | 85.1(0.2) | 19.0(3.3) | 13.3(2.8) | 83.0(0.3) | 27.8(1.5) | 6.7(0.9) | 59.5(2.3) | 17.1(3.4) | 10.1(3.0) |
| InfoCORE | **85.2(0.1)** | **14.9(1.1)** | **9.7(0.8)** | 82.7(0.4) | **25.4(3.6)** | **6.0(1.4)** | 60.1(2.1) | **15.3(2.5)** | **9.3(0.8)** |

cion & Newman, 2007), Law School (Wightman, 1998), and Compas (Angwin et al., 2022). Race and gender are used as protected attributes, separating the data into four subgroups. Considering that minority groups often encounter data limitation issues, we subsample the training data to mimic an unbalanced population. To assess representation fairness, we follow the setup in Lahoti et al. (2020) and Ma et al. (2021) and analyze three common fairness criteria (Feldman et al., 2015; Hardt

et al., 2016): equalized odds (EO), equality of opportunity (EOPP), and demographic parity (DP), with definitions in Appendix G and DP results in Appendix I. We also calculate prediction accuracy (Acc) to account for the utility-fairness trade-off in the representations (Zhao & Gordon, 2022).

**Results.** As shown in Table 4, while achieving similar levels of prediction accuracy, InfoCORE consistently obtains more fair representations across datasets in terms of the different criteria, especially in UCI Adult. CCL improves representation fairness over CLIP by constraining the negative sampling to be within the same sensitive attribute subgroup. However, given the limited sample size of the minority groups, it performs inferior to InfoCORE.

## 4 RELATED WORK

**Representation Learning for Molecules.** Traditional unimodal techniques for molecular representation learning (Rogers & Hahn, 2010; Durant et al., 2002; Wang et al., 2022; Xu et al., 2021b; Wang et al., 2019) often falter because molecules with similar structures can have very different effects in the cellular context. Given such limitations, researchers are increasingly turning to multimodal methods – incorporating additional modalities like 3D structure (Stärk et al., 2022; Zhou et al., 2022) as well as high-throughput cell imaging. For instance, Nguyen et al. (2023) utilize the CLIP model (Radford et al., 2021) to learn multimodal molecular and cell image representations. Zheng et al. (2022) incorporate masked graph modeling and generative graph-image matching objectives to elevate the quality of the learned representations. However, they have difficulties generalizing across molecular structures due to the variability and batch effect in the drug screens as the training data, which can introduce confounding variables and bias the representations.

**Batch Effect Removal.** Various approaches have been proposed to correct for batch effects, including a linear method (Johnson et al., 2007), a mixture-model based method (Korsunsky et al., 2019), neighbor-based methods (Hie et al., 2019; Haghverdi et al., 2018), and variational-inference based methods (Lopez et al., 2018; Li et al., 2020). These methods, typically used as independent preprocessing steps designed for specific biology experimental techniques, are challenging to be applied to more general contexts. In contrast, our approach treats batch effects as sensitive attributes to remove, providing a flexible framework applicable across various domains.

**Representation Learning with Sensitive Attributes.** Wu et al. (2022); Chen et al. (2022); Yang et al. (2022); Miao et al. (2022); Fan et al. (2022) learn invariant features for better out-of distribution generalization in graph classification or interpretability. However, they focus on the unimodal supervised learning setting. In the unsupervised setting, to learn fair representations, Song et al. (2019) use variational and adversarial objectives as a tractable lower bound on conditional mutual information, which requires a challenging tri-level optimization. Ma et al. (2021) relate conditional mutual information with a conditional contrastive learning objective, where the negative pairs are drawn from the conditional marginal distributions. Tsai et al. (2021) further extend it to handle continuous conditioning variables by leveraging similarity kernels. However, this does not resolve the confounder issue in drug screening data, since observations per batch are limited and similarity between batches as a categorical identifier is hard to measure. Zhang et al. (2022) rely on recent advances in image generation to obtain balanced training data prior to the representation learning task, but generative models for high-content screening data are in their infancy (Yang et al., 2021a).

## 5 CONCLUSION

We present InfoCORE, a general framework for multimodal molecular representation learning. InfoCORE effectively integrates diverse high-content drug screens with the chemical structure in the presence of confounders like batch effects. Theoretically, we show that InfoCORE maximizes the variational lower bound on the conditional mutual information of the representations given the batch identifier. Empirically, we demonstrate that InfoCORE outperforms baselines on two drug screening datasets (gene expression and single-cell imaging) on two tasks, molecular property prediction and molecule-phenotype retrieval. Finally, we show that the information maximization principle underlying InfoCORE extends well beyond drug discovery and can be viewed as a general-purpose framework. In particular, we empirically demonstrate its efficacy in eliminating sensitive information, focusing on fairness applications across three major datasets and various fairness criteria.

ACKNOWLEDGMENTS

We thank Adityanarayanan Radhakrishnan, Joshua Robinson, Yonglong Tian, Yilun Xu, Shangyuan Tong, Wengong Jin, Hannes Stärk, Boyuan Chen, Zongyu Lin, and Mengfei Xia for helpful discussions and comments on the manuscript.

CW and TJ acknowledge support from the Machine Learning for Pharmaceutical Discovery and Synthesis (MLPDS) consortium. SG acknowledges funding from the Office of Naval Research grant N00014-20-1-2023 (MURI ML-SCOPE) and NSF award CCF-2112665 (TILOS AI Institute). CU acknowledges support by NCCIH/NIH (1DP2AT012345), ONR (N00014-22-1-2116), the MIT-IBM Watson AI Lab, AstraZeneca, the Eric and Wendy Schmidt Center at the Broad Institute, and a Simons Investigator Award.

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

## A    PROOF OF PROPOSITION 1.

**Proposition 1.** *Given random variables $Z_g, Z_d$ as representations of two data modalities and $X_b$ as the irrelevant attribute, the conditional mutual information $I(Z_d; Z_g|X_b)$ between representations conditioned on the irrelevant attribute has the following lower bound:*

$$I(Z_d; Z_g|X_b) \geq \mathbb{E}_{p(z_d,z_g,x_b)}\left[h(z_d, z_g, x_b)\right] - e^{-1}\mathbb{E}_{p(z_d)}\mathbb{E}_{p(z_g,x_b)}\left[e^{h(z_d,z_g,x_b)}\right] - I(Z_d; X_b), \quad (5)$$

*which results from using an energy-based variational family $q(z_g, x_b|z_d)$ to approximate the distribution $p(z_g, x_b|z_d)$:*

$$q(z_g, x_b|z_d) = \frac{p(z_g, x_b)}{Z(z_d)}e^{h(z_d,z_g,x_b)}, \text{ where } Z(z_d) = \mathbb{E}_{p(z_g,x_b)}\left[e^{h(z_d,z_g,x_b)}\right] \text{ is the partition function,}$$

*with equality holding when the critic $h$ satisfies $h^*(z_d, z_g, x_b) = 1 + \log\frac{p(z_g,x_b|z_d)}{p(z_g,x_b)}$.*

*Proof.* By definition of conditional mutual information,

$$I(Z_d; Z_g|X_b) = \mathbb{E}_{p(z_g,z_d,x_b)}\left[\log\frac{p(x_b)\cdot p(z_g,x_b|z_d)}{p(x_b|z_d)\cdot p(z_g,x_b)}\right].$$

Replace the intractable conditional distribution $p(z_g, x_b|z_d)$ with the corresponding variational distribution $q(z_g, x_b|z_d)$, we can get a lower bound due to the non-negativity of KL divergence:

$$I(Z_d; Z_g|X_b) = \mathbb{E}_{p(z_g,z_d,x_b)}\left[\log\frac{p(x_b)\cdot q(z_g,x_b|z_d)}{p(x_b|z_d)\cdot p(z_g,x_b)}\right] + \mathbb{E}_{p(z_d)}\left[D_{\text{KL}}\left(p\left(z_g,x_b|z_d\right)\|q\left(z_g,x_b|z_d\right)\right)\right]$$

$$\geq \mathbb{E}_{p(z_g,z_d,x_b)}\left[\log\frac{p(x_b)\cdot q(z_g,x_b|z_d)}{p(x_b|z_d)\cdot p(z_g,x_b)}\right].$$

$$(6)$$

We can choose an energy-based variational family that uses a critic $h(z_d, z_g, x_b)$, scaled by the marginal density $p(z_g, x_b)$ and normalized by the partition function $Z(z_d)$:

$$q(z_g, x_b|z_d) = \frac{p(z_g, x_b)}{Z(z_d)}e^{h(z_d,z_g,x_b)}, \text{ where } Z(z_d) = \mathbb{E}_{p(z_g,x_b)}\left[e^{h(z_d,z_g,x_b)}\right].$$

By plugging this into the lower bound in Equation 6, we obtain

$$I(Z_d; Z_g|X_b) \geq \mathbb{E}_{p(z_g,z_d,x_b)}\left[\log\frac{p(x_b)\cdot e^{h(z_d,z_g,x_b)}}{p(x_b|z_d)\cdot Z(z_d)}\right]$$

$$= \mathbb{E}_{p(z_d,z_g,x_b)}\left[h(z_d, z_g, x_b)\right] - \mathbb{E}_{p(z_d)}\left[\log Z(z_d)\right] - I(Z_d; X_b).$$

The intractable log-partition function can be upper bounded using the inequality $\log(x) \leq \frac{x}{a} + \log(a) - 1, \forall x, a > 0$, which is tight when $x = a$. Applying the inequality to $\log Z(z_d)$ and taking $a = e$, we obtain the following upper bound of the log-partition function:

$$\log Z(z_d) \leq e^{-1}Z(z_d). \quad (7)$$

This results in the following single sample lower bound of the conditional mutual information:

$$I(Z_d; Z_g|X_b) \geq \mathbb{E}_{p(z_d,z_g,x_b)}\left[h(z_d, z_g, x_b)\right] - e^{-1}\mathbb{E}_{p(z_d)}\mathbb{E}_{p(z_g,x_b)}\left[e^{h(z_d,z_g,x_b)}\right] - I(Z_d; X_b).$$

Denoting the optimal critic as $h^*(z_d, z_g, x_b)$, Equation 6 is tight when $q(z_g, x_b|z_d) = p(z_g, x_b|z_d)$, i.e.

$$h^*(z_d, z_g, x_b) = \log p(z_g, x_b|z_d) + c(z_g, x_b),$$

where $c(z_d, x_b)$ is an arbitrary function solely depend on $z_d$ and $x_b$.

Equation 7 is tight when $Z(z_d) = e$, i.e. ,

$$e = \int_{z_g,x_b} p(z_g, x_b)e^{h^*(z_d,z_g,x_b)}\,\mathrm{d}z_g\mathrm{d}x_b = \int_{z_g,x_b} p(z_g, x_b)e^{c(z_g,x_b)}p(z_g, x_b|z_d)\,\mathrm{d}z_g\mathrm{d}x_b,$$

which holds when $c(z_g, x_b) = 1 - \log p(z_g, x_b)$.

Thus, the optimal critic satisfies $h^*(z_d, z_g, x_b) = 1 + \log\frac{p(z_g,x_b|z_d)}{p(z_g,x_b)}$, which completes the proof. $\quad\square$

## B  PROOF OF PROPOSITION 2.

**Proposition 2.** *Given samples $(z_d^1, z_g^1, x_b^1)$ drawn from the joint distribution $(Z_d^1, Z_g^1, X_b^1) \sim p(z_d, z_g, x_b)$ and $z_d^{2:K}$ drawn i.i.d. from the marginal distribution $Z_d^i \sim p(z_d)$ for $i = 2, ..., K$, then the conditional mutual information $I(Z_d^1; Z_g^1 | X_b^1)$ has the following lower bound:*

$$I(Z_d^1; Z_g^1 | X_b^1) \geq -L_{CLIP} - L_{CLF} + C - H(X_b^1), \tag{8}$$

*where* $L_{CLIP} = -\dfrac{1}{2}\left[ \mathbb{E}_{p(z_d^1, z_g^1, x_b^1)p(z_d^{2:K})}\left[ \log \dfrac{e^{f(z_g^1, z_d^1)}}{\frac{1}{K}\sum_{i=1}^K e^{f(z_g^1, z_d^i)} \cdot \hat{p}_g(x_b^1 | z_g^1, z_d^i)} \right]\right.$

$\left. + \mathbb{E}_{p(z_d^1, z_g^1, x_b^1)p(z_g^{2:K})}\left[ \log \dfrac{e^{f(z_g^1, z_d^1)}}{\frac{1}{K}\sum_{i=1}^K e^{f(z_g^i, z_d^1)} \cdot \hat{p}_d(x_b^1 | z_g^i, z_d^1)} \right]\right],$

$L_{CLF} = \dfrac{1}{2}\left[ \mathbb{E}_{p(z_d^1)}\left[ D_{\text{KL}}\left( p\left(x_b^1 | z_d^1\right) \| \hat{p}\left(x_b^1 | z_d^1\right)\right)\right] + \mathbb{E}_{p(z_g^1)}\left[ D_{\text{KL}}\left( p\left(x_b^1 | z_g^1\right) \| \hat{p}\left(x_b^1 | z_g^1\right)\right)\right]\right],$

$C = \dfrac{1}{2}\,\mathbb{E}_{p(z_d^1, z_g^1, x_b^1)}\left[ \log \dfrac{\hat{p}_g(x_b^1 | z_g^1, z_d^1) \cdot \hat{p}_d(x_b^1 | z_g^1, z_d^1)}{\hat{p}(x_b^1 | z_g^1) \cdot \hat{p}(x_b^1 | z_d^1)} \right].$

*The lower bound holds for any choice of critic $f(z_g, z_d)$ and variational distribution $\hat{p}_g(x_b | z_g, z_d)$, $\hat{p}_d(x_b | z_g, z_d)$, with equality holding when $f^*(z_d, z_g) = \log \frac{p(z_g | z_d)}{p(z_g)}$ and $\hat{p}_g^*(x_b | z_g, z_d) = \hat{p}_d^*(x_b | z_g, z_d) = p(x_b | z_g, z_d)$.*[4]

*Proof.* By definition of conditional mutual information,

$$I(Z_d^1; Z_g^1 | X_b^1) = I(Z_d^{1:K}; Z_g^1 | X_b^1) = \mathbb{E}_{p(z_g^1, z_d^1, x_b^1)p(z_d^{2:K})}\left[ \log \frac{p(x_b^1) \cdot p(z_d^1, z_g^1, x_b^1)}{p(z_d^1, x_b^1) \cdot p(z_g^1, x_b^1)} \right]$$

$$= \mathbb{E}_{p(z_g^1, z_d^1, x_b^1)p(z_d^{2:K})}\left[ \log \frac{\frac{p(x_b^1)}{p(x_b^1 | z_d^1)} \cdot p(z_g^1, x_b^1 | z_d^1)}{p(z_g^1, x_b^1)} \right].$$

Then applying a variational distribution $q(z_g^1, x_b^1 | z_d^{1:K})$ to approximate $p(z_g^1, x_b^1 | z_d^{1:K}) = p(z_g^1, x_b^1 | z_d^1)$ yields the following lower bound on the conditional mutual information:

$$I(Z_d^1; Z_g^1 | X_b^1) \geq \mathbb{E}_{p(z_d^1, z_g^1, x_b^1)p(z_d^{2:K})}\left[ \log \frac{\frac{p(x_b^1)}{p(x_b^1 | z_d^1)} \cdot q(z_g^1, x_b^1 | z_d^{1:K})}{p(z_g^1, x_b^1)} \right].$$

Observing that $p(z_g^1, x_b^1 | z_d^1) = p(z_g^1 | z_d^1) \cdot p(x_b^1 | z_g^1, z_d^1)$, we can approximate the two terms separately. For this, we define the variational distribution as:

$$q(z_g^1, x_b^1 | z_d^{1:K}) = \frac{e \cdot p(z_g^1, x_b^1) \frac{e^{f(z_d^1, z_g^1)}\hat{p}_g(x_b^1 | z_g^1, z_d^1)}{a(z_g^1, x_b^1; z_d^{1:K})}}{Z(z_d^{1:K})},$$

$$\text{where } Z(z_d^{1:K}) = e \cdot \mathbb{E}_{p(z_g^1, x_b^1)}\left[ \frac{e^{f(z_d^1, z_g^1)}\hat{p}_g(x_b^1 | z_g^1, z_d^1)}{a(z_g^1, x_b^1; z_d^{1:K})} \right],$$

$$a(z_g^1, x_b^1; z_d^{1:K}) = \frac{1}{K}\sum_{i=1}^K e^{f(z_d^i, z_g)}\hat{p}_g(x_b^1 | z_g^1, z_d^i)$$

---

[4]We note that $\hat{p}(x_b | z_g)$ and $\hat{p}(x_b | z_d)$ cancel out in $-L_{\text{CLF}} + C$, but we spell out these terms for ease of estimating $C$ later.

By plugging this term into the inequality above, we obtain

$$I(Z_d^1; Z_g^1 | X_b^1) \geq \mathbb{E}_{p(z_d^1, z_g^1, x_b^1)p(z_d^{2:K})} \left[ \log \frac{e \cdot \frac{p(x_b^1)}{p(x_b^1|z_d^1)} \cdot \frac{e^{f(z_d^1, z_g^1)} \hat{p}_g(x_b^1|z_g^1, z_d^1)}{a(z_g^1, x_b^1; z_d^{1:K})}}{Z(z_d^{1:K})} \right]$$

$$= 1 + \mathbb{E}_{p(z_d^1, z_g^1, x_b^1)p(z_d^{2:K})} \left[ \log \frac{\frac{p(x_b^1)}{p(x_b^1|z_d^1)} \cdot e^{f(z_d^1, z_g^1)} \hat{p}_g(x_b^1|z_g^1, z_d^1)}{a(z_g^1, x_b^1; z_d^{1:K})} \right] - \mathbb{E}_{p(z_d^{1:K})} \left[ \log Z(z_d^{1:K}) \right]$$

$$\geq 1 + \mathbb{E}_{p(z_d^1, z_g^1, x_b^1)p(z_d^{2:K})} \left[ \log \frac{\frac{p(x_b^1)}{p(x_b^1|z_d^1)} \cdot e^{f(z_d^1, z_g^1)} \hat{p}_g(x_b^1|z_g^1, z_d^1)}{a(z_g^1, x_b^1; z_d^{1:K})} \right] - e^{-1} \mathbb{E}_{p(z_d^{1:K})} \left[ Z(z_d^{1:K}) \right].$$

For the last inequality we used $\log x \leq \frac{x}{a} + \log a - 1$ and took $a = e$.

By symmetry, we can conclude as follows that the last term equals to the constant 1:

$$e^{-1} \mathbb{E}_{p(z_d^{1:K})} \left[ Z(z_d^{1:K}) \right] = e^{-1} \mathbb{E}_{p(z_d^{1:K})} \left[ e \cdot \mathbb{E}_{p(z_g^1, x_b^1)} \left[ \frac{e^{f(z_d^1, z_g^1)} \hat{p}_g(x_b^1|z_g^1, z_d^1)}{a(z_g^1, x_b^1; z_d^{1:K})} \right] \right]$$

$$= \mathbb{E}_{p(z_g^1, x_b^1)p(z_d^{1:K})} \left[ \frac{e^{f(z_d^1, z_g^1)} \hat{p}_g(x_b^1|z_g^1, z_d^1)}{a(z_g^1, x_b^1; z_d^{1:K})} \right]$$

$$= \frac{1}{K} \sum_{i=1}^{K} \mathbb{E}_{p(z_g^1, x_b^1)p(z_d^{1:K})} \left[ \frac{e^{f(z_d^i, z_g)} \hat{p}_g(x_b^1|z_g^1, z_d^i)}{a(z_g^1, x_b^1; z_d^{1:K})} \right]$$

$$= \mathbb{E}_{p(z_g^1, x_b^1)p(z_d^{1:K})} \left[ \frac{\frac{1}{K} \sum_{i=1}^{K} e^{f(z_d^i, z_g)} \hat{p}_g(x_b^1|z_g^1, z_d^i)}{a(z_g^1, x_b^1; z_d^{1:K})} \right] = 1.$$

Therefore,

$$I(Z_d^1; Z_g^1 | X_b^1) = I(Z_d^{1:K}; Z_g^1 | X_b^1) \geq \mathbb{E}_{p(z_d^1, z_g^1, x_b^1)p(z_d^{2:K})} \left[ \log \frac{\frac{p(x_b^1)}{p(x_b^1|z_d^1)} \cdot e^{f(z_d^1, z_g^1)} \hat{p}_g(x_b^1|z_g^1, z_d^1)}{a(z_g^1, x_b^1; z_d^{1:K})} \right]$$

with optimal critics $f^*(z_d, z_g) = \log p(z_g|z_d) + c(z_g)$; $\hat{p}_g^*(x_b|z_g, z_d) = p(x_b|z_g, z_d)$.

Using $\hat{p}(x_b^1|z_d^1)$ as an estimator of the posterior batch distribution $p(x_b^1|z_d^1)$, the lower bound can be written as:

$$I(Z_d^1; Z_g^1 | X_b^1) \geq \mathbb{E}_{p(z_d^1, z_g^1, x_b^1)p(z_d^{2:K})} \left[ \log \frac{\frac{p(x_b^1)}{\hat{p}(x_b^1|z_d^1)} e^{f(z_g^1, z_d^1)} \cdot \hat{p}_g(x_b^1|z_g^1, z_d^1)}{\frac{1}{K} \sum_{i=1}^{K} e^{f(z_g^1, z_d^i)} \cdot \hat{p}_g(x_b^1|z_g^1, z_d^i)} \cdot \frac{\hat{p}(x_b^1|z_d^1)}{p(x_b^1|z_d^1)} \right]$$

$$= \mathbb{E}_{p(z_d^1, z_g^1, x_b^1)p(z_d^{2:K})} \left[ \log \frac{\frac{p(x_b^1)}{\hat{p}(x_b^1|z_d^1)} e^{f(z_g^1, z_d^1)} \cdot \hat{p}_g(x_b^1|z_g^1, z_d^1)}{\frac{1}{K} \sum_{i=1}^{K} e^{f(z_g^1, z_d^i)} \cdot \hat{p}_g(x_b^1|z_g^1, z_d^i)} \right] - \mathbb{E}_{p(z_d^1, x_b^1)} \left[ \log \frac{p(x_b^1|z_d^1)}{\hat{p}(x_b^1|z_d^1)} \right].$$

Note that the last term is the expectation of the KL-divergence between $p(x_b^1|z_d^1)$ and $\hat{p}(x_b^1|z_d^1)$, i.e.,

$$\mathbb{E}_{p(z_d^1, x_b^1)} \left[ \log \frac{p(x_b^1|z_d^1)}{\hat{p}(x_b^1|z_d^1)} \right] = \mathbb{E}_{p(z_d^1)} \left[ \mathbb{E}_{p(x_b^1|z_d^1)} \left[ \log \frac{p(x_b^1|z_d^1)}{\hat{p}(x_b^1|z_d^1)} \right] \right] = \mathbb{E}_{p(z_d^1)} \left[ D_{\mathrm{KL}} \left( p\left(x_b^1|z_d^1\right) \| \hat{p}\left(x_b^1|z_d^1\right) \right) \right].$$

Thus the lower bound can be rewritten as follows:

$$
I(Z_d^1; Z_g^1 | X_b^1) \geq \mathbb{E}_{p(z_d^1, z_g^1, x_b^1) p(z_d^{2:K})} \left[ \log \frac{\frac{p(x_b^1)}{\hat{p}(x_b^1 | z_d^1)} e^{f(z_g^1, z_d^1)} \cdot \hat{p}_g(x_b^1 | z_g^1, z_d^1)}{\frac{1}{K} \sum_{i=1}^{K} e^{f(z_g^1, z_d^i)} \cdot \hat{p}_g(x_b^1 | z_g^1, z_d^i)} \right]
$$

$$
- \mathbb{E}_{p(z_d^1)} \left[ D_{\mathrm{KL}} \left( p\left(x_b^1 | z_d^1\right) \| \hat{p}\left(x_b^1 | z_d^1\right) \right) \right]
$$

$$
= \mathbb{E}_{p(z_d^1, z_g^1, x_b^1) p(z_d^{2:K})} \left[ \log \frac{e^{f(z_g^1, z_d^1)}}{\frac{1}{K} \sum_{i=1}^{K} e^{f(z_g^1, z_d^i)} \cdot \hat{p}_g(x_b^1 | z_g^1, z_d^i)} \right]
$$

$$
+ \mathbb{E}_{p(z_d^1, z_g^1, x_b^1)} \left[ \log \frac{\hat{p}_g(x_b^1 | z_g^1, z_d^1)}{\hat{p}(x_b^1 | z_d^1)} \right] - H(X_b^1) - \mathbb{E}_{p(z_d^1)} \left[ D_{\mathrm{KL}} \left( p\left(x_b^1 | z_d^1\right) \| \hat{p}\left(x_b^1 | z_d^1\right) \right) \right].
$$

Note that, anagolously, we have

$$
I(Z_d^1; Z_g^1 | X_b^1) = I(Z_d^1; Z_g^{1:K} | X_b^1)
$$

$$
\geq \mathbb{E}_{p(z_d^1, z_g^1, x_b^1) p(z_g^{2:K})} \left[ \log \frac{e^{f(z_g^1, z_d^1)}}{\frac{1}{K} \sum_{i=1}^{K} e^{f(z_g^i, z_d^1)} \cdot \hat{p}_d(x_b^1 | z_g^i, z_d^1)} \right]
$$

$$
+ \mathbb{E}_{p(z_d^1, z_g^1, x_b^1)} \left[ \log \frac{\hat{p}_d(x_b^1 | z_g^1, z_d^1)}{\hat{p}(x_b^1 | z_g^1)} \right] - H(X_b^1) - \mathbb{E}_{p(z_g^1)} \left[ D_{\mathrm{KL}} \left( p\left(x_b^1 | z_g^1\right) \| \hat{p}\left(x_b^1 | z_g^1\right) \right) \right]
$$

with optimal critics $f^*(z_d, z_g) = \log p(z_d | z_g) + c(z_d)$; $\hat{p}_d^*(x_b | z_g, z_d) = p(x_b | z_g, z_d)$.

Incorporating the two losses, we obtain

$$
I(Z_d^1; Z_g^1 | X_b^1) = \frac{1}{2} \left[ I(Z_d^{1:K}; Z_g^1 | X_b^1) + I(Z_d^1; Z_g^{1:K} | X_b^1) \right]
$$

$$
\geq \frac{1}{2} \left[ \mathbb{E}_{p(z_d^1, z_g^1, x_b^1) p(z_d^{2:K})} \left[ \log \frac{e^{f(z_g^1, z_d^1)}}{\frac{1}{K} \sum_{i=1}^{K} e^{f(z_g^1, z_d^i)} \cdot \hat{p}_g(x_b^1 | z_g^1, z_d^i)} \right] \right.
$$

$$
\left. + \mathbb{E}_{p(z_d^1, z_g^1, x_b^1) p(z_g^{2:K})} \left[ \log \frac{e^{f(z_g^1, z_d^1)}}{\frac{1}{K} \sum_{i=1}^{K} e^{f(z_g^i, z_d^1)} \cdot \hat{p}_d(x_b^1 | z_g^i, z_d^1)} \right] \right]
$$

$$
- \frac{1}{2} \left[ \mathbb{E}_{p(z_d^1)} \left[ D_{\mathrm{KL}} \left( p\left(x_b^1 | z_d^1\right) \| \hat{p}\left(x_b^1 | z_d^1\right) \right) \right] + \mathbb{E}_{p(z_g^1)} \left[ D_{\mathrm{KL}} \left( p\left(x_b^1 | z_g^1\right) \| \hat{p}\left(x_b^1 | z_g^1\right) \right) \right] \right]
$$

$$
+ \frac{1}{2} \mathbb{E}_{p(z_d^1, z_g^1, x_b^1)} \left[ \log \frac{\hat{p}_g(x_b^1 | z_g^1, z_d^1) \cdot \hat{p}_d(x_b^1 | z_g^1, z_d^1)}{\hat{p}(x_b^1 | z_g^1) \cdot \hat{p}(x_b^1 | z_d^1)} \right] - H(X_b^1).
$$

When $c(z_g) = -\log p(z_g)$, and $c(z_d) = -\log p(z_d)$, the two optimal critics agree with each other, i.e., $f^*(z_d, z_g) = \log \frac{p(z_g | z_d)}{p(z_g)} = \log \frac{p(z_d | z_g)}{p(z_d)}$, $\hat{p}_g^*(x_b | z_g, z_d) = \hat{p}_d^*(x_b | z_g, z_d) = p(x_b | z_g, z_d)$, which completes the proof. $\square$

Note that the KL-divergence in $L_{\mathrm{CLF}}$ can be equivalently written as the sum of an entropy term and a cross-entropy term:

$$
D_{\mathrm{KL}} \left( p\left(x_b^1 | z_d^1\right) \| \hat{p}\left(x_b^1 | z_d^1\right) \right) = H\left( p\left(x_b^1 | z_d^1\right) \right) - \mathbb{E}_{p(x_b^1 | z_d^1)} \left[ \log \hat{p}(x_b^1 | z_d^1) \right]
$$

$$
= H\left( p\left(x_b^1 | z_d^1\right) \right) + CE\left( p(x_b^1 | z_d^1), \hat{p}(x_b^1 | z_d^1) \right),
$$

$$
D_{\mathrm{KL}} \left( p\left(x_b^1 | z_g^1\right) \| \hat{p}\left(x_b^1 | z_g^1\right) \right) = H\left( p\left(x_b^1 | z_g^1\right) \right) - \mathbb{E}_{p(x_b^1 | z_g^1)} \left[ \log \hat{p}(x_b^1 | z_g^1) \right]
$$

$$
= H\left( p\left(x_b^1 | z_g^1\right) \right) + CE\left( p(x_b^1 | z_g^1), \hat{p}(x_b^1 | z_g^1) \right).
$$

Then $L_{\mathrm{CLF}}$ can be factorized into the cross-entropy loss of the classifiers $\hat{p}(x_b^1 | z_d^1)$ and $\hat{p}(x_b^1 | z_g^1)$, and a constant term. Therefore, minimizing $L_{\mathrm{CLF}}$ can be achieved by optimizing the classifiers via the cross-entropy loss.

## C  PROOF OF PROPOSITION 3

**Proposition 3.** *When estimating $\hat{p}_g(x_b^1|z_g^1, z_d^i)$ as the weighted average of $\hat{p}(x_b^1|z_g^1)$ and $\hat{p}(x_b^1|z_d^i)$, and analogously for $\hat{p}_d(x_b^1|z_g^i, z_d^1)$, i.e.*

$$\hat{p}_g(x_b^1|z_g^1, z_d^i) = \alpha \cdot \hat{p}(x_b^1|z_g^1) + (1-\alpha) \cdot \hat{p}(x_b^1|z_d^i),\ \hat{p}_d(x_b^1|z_g^i, z_d^1) = \alpha \cdot \hat{p}(x_b^1|z_d^1) + (1-\alpha) \cdot \hat{p}(x_b^1|z_g^i),$$

*then $C = \frac{1}{2} \mathbb{E}_{p(z_d^1, z_g^1, x_b^1)} \left[ \log \frac{\hat{p}_g(x_b^1|z_g^1, z_d^1) \cdot \hat{p}_d(x_b^1|z_g^1, z_d^1)}{\hat{p}(x_b^1|z_g^1) \cdot \hat{p}(x_b^1|z_d^1)} \right]$ is lower bounded by the constant zero.*

*Proof.* The weighted AM-GM inequality indicates that for non-negative numbers $\{x_i\}_{i=1}^n$ and non-negative weights $\{w_i\}_{i=1}^n$, we have the inequality

$$\frac{w_1 x_1 + w_2 x_2 + \cdots + w_n x_n}{w} \geq \sqrt[w]{x_1^{w_1} x_2^{w_2} \cdots x_n^{w_n}},$$

where $w = w_1 + w_2 + \cdots + w_n$.

Thus, by applying the weighted AM-GM inequality, we obtain

$$
\begin{aligned}
C &= \mathbb{E}_{p(z_d^1, z_g^1, x_b^1)} \left[ \log \frac{\hat{p}_g(x_b^1|z_g^1, z_d^1) \cdot \hat{p}_d(x_b^1|z_g^1, z_d^1)}{\hat{p}(x_b^1|z_g^1) \cdot \hat{p}(x_b^1|z_d^1)} \right] \\
&= \mathbb{E}_{p(z_d^1, z_g^1, x_b^1)} \left[ \log \frac{\left(\alpha \cdot \hat{p}(x_b^1|z_g^1) + (1-\alpha) \cdot \hat{p}(x_b^1|z_d^1)\right) \cdot \left(\alpha \cdot \hat{p}(x_b^1|z_d^1) + (1-\alpha) \cdot \hat{p}(x_b^1|z_g^1)\right)}{\hat{p}(x_b^1|z_g^1) \cdot \hat{p}(x_b^1|z_d^1)} \right] \\
&\geq \mathbb{E}_{p(z_d^1, z_g^1, x_b^1)} \left[ \log \frac{\left(\hat{p}(x_b^1|z_g^1)^\alpha \cdot \hat{p}(x_b^1|z_d^1)^{1-\alpha}\right) \cdot \left(\hat{p}(x_b^1|z_d^1)^\alpha \cdot \hat{p}(x_b^1|z_g^1)^{1-\alpha}\right)}{\hat{p}(x_b^1|z_g^1) \cdot \hat{p}(x_b^1|z_d^1)} \right] \\
&= \mathbb{E}_{p(z_d^1, z_g^1, x_b^1)} \log 1 = 0,
\end{aligned}
$$

which completes the proof. $\square$

Additionally, we show that this bound can be generalized from the arithmetic average to the weighted geometric average, i.e.,

$$\hat{p}_g(x_b^1|z_g^1, z_d^i) \propto \hat{p}(x_b^1|z_g^1)^\alpha \cdot \hat{p}(x_b^1|z_d^i)^{1-\alpha},\ \hat{p}_d(x_b^1|z_g^i, z_d^1) \propto \hat{p}(x_b^1|z_d^1)^\alpha \cdot \hat{p}(x_b^1|z_g^i)^{1-\alpha}.$$

In this case, the estimated probabilities need to be normalized:

$$
\begin{aligned}
\hat{p}_g(x_b^1|z_g^1, z_d^i) &= \frac{\hat{p}(x_b^1|z_g^1)^\alpha \cdot \hat{p}(x_b^1|z_d^i)^{1-\alpha}}{\sum_{x_b'} \hat{p}(x_b'|z_g^1)^\alpha \cdot \hat{p}(x_b'|z_d^i)^{1-\alpha}}, \\
\hat{p}_d(x_b^1|z_g^i, z_d^1) &= \frac{\hat{p}(x_b^1|z_d^1)^\alpha \cdot \hat{p}(x_b^1|z_g^i)^{1-\alpha}}{\sum_{x_b'} \hat{p}(x_b'|z_d^1)^\alpha \cdot \hat{p}(x_b'|z_g^i)^{1-\alpha}}.
\end{aligned}
$$

Applying the weighted AM-GM inequality, we obtain

$$
\begin{aligned}
\sum_{x_b'} \hat{p}(x_b'|z_g^1)^\alpha \cdot \hat{p}(x_b'|z_d^1)^{1-\alpha} &\leq \sum_{x_b'} \alpha \cdot \hat{p}(x_b'|z_g^1) + (1-\alpha) \cdot \hat{p}(x_b'|z_d^1) \\
&= \alpha \cdot \sum_{x_b'} \hat{p}(x_b'|z_g^1) + (1-\alpha) \sum_{x_b'} \hat{p}(x_b'|z_d^1) = 1.
\end{aligned}
$$

Analogously, we obtain

$$\sum_{x_b'} \hat{p}(x_b'|z_d^1)^\alpha \cdot \hat{p}(x_b'|z_g^1)^{1-\alpha} \leq 1.$$

Plugging this into the definition of $C$, we obtain

$$C = \mathbb{E}_{p(z_d^1, z_g^1, x_b^1)} \left[ \log \frac{\hat{p}_g(x_b^1 | z_g^1, z_d^1) \cdot \hat{p}_d(x_b^1 | z_g^1, z_d^1)}{\hat{p}(x_b^1 | z_g^1) \cdot \hat{p}(x_b^1 | z_d^1)} \right]$$

$$= \mathbb{E}_{p(z_d^1, z_g^1, x_b^1)} \left[ \log \frac{\frac{\hat{p}(x_b^1 | z_g^1)^\alpha \cdot \hat{p}(x_b^1 | z_d^1)^{1-\alpha}}{\sum_{x_b'} \hat{p}(x_b' | z_g^1)^\alpha \cdot \hat{p}(x_b' | z_d^1)^{1-\alpha}} \cdot \frac{\hat{p}(x_b^1 | z_d^1)^\alpha \cdot \hat{p}(x_b^1 | z_g^1)^{1-\alpha}}{\sum_{x_b'} \hat{p}(x_b' | z_d^1)^\alpha \cdot \hat{p}(x_b' | z_g^1)^{1-\alpha}}}{\hat{p}(x_b^1 | z_g^1) \cdot \hat{p}(x_b^1 | z_d^1)} \right]$$

$$= -\mathbb{E}_{p(z_d^1, z_g^1, x_b^1)} \left[ \log \sum_{x_b'} \hat{p}(x_b' | z_g^1)^\alpha \cdot \hat{p}(x_b' | z_d^1)^{1-\alpha} \cdot \sum_{x_b'} \hat{p}(x_b' | z_d^1)^\alpha \cdot \hat{p}(x_b' | z_g^1)^{1-\alpha} \right] \geq 0.$$

Therefore, $C$ is lower bounded by the constant zero if $\hat{p}(x_b^1 | z_g^1, z_d^i)$ (and analogously for $\hat{p}(x_b^1 | z_g^i, z_d^1)$) is estimated as the weighted (either arithmetic or geometric) average of $\hat{p}(x_b^1 | z_g^1)$ and $\hat{p}(x_b^1 | z_d^i)$.

# D    GRADIENT OF $L_{\text{CLIP}}$ W.R.T. REPRESENTATIONS AND OPTIMIZATION DETAILS.

Denote

$$L_{\text{CLIP}}^g = -\mathbb{E}_{p(z_d^1, z_g^1, x_b^1)p(z_d^{2:K})} \left[ \log \frac{e^{f(z_g^1, z_d^1)}}{\frac{1}{K} \sum_{i=1}^K e^{f(z_g^1, z_d^i)} \cdot \left( \alpha \cdot \hat{p}(x_b^1 | z_g^1) + (1 - \alpha) \cdot \hat{p}(x_b^1 | z_d^i) \right)} \right],$$

$$L_{\text{CLIP}}^t = -\mathbb{E}_{p(z_d^1, z_g^1, x_b^1)p(z_g^{2:K})} \left[ \log \frac{e^{f(z_g^1, z_d^1)}}{\frac{1}{K} \sum_{i=1}^K e^{f(z_g^i, z_d^1)} \cdot \left( \alpha \cdot \hat{p}(x_b^1 | z_d^1) + (1 - \alpha) \cdot \hat{p}(x_b^1 | z_g^i) \right)} \right].$$

Note that the gradients of $L_{\text{CLIP}}^g$ w.r.t. anchor $z_g^1$, positive $z_d^1$, and negatives $z_d^i$ ($i \neq 1$) have the following form:

$$\frac{\partial L_{\text{CLIP}}^g}{\partial z_g^1} = \mathbb{E}_{p(z_d^1, z_g^1, x_b^1)p(z_d^{2:K})} \left[ \frac{\sum_{i=1}^K e^{f(z_g^1, z_d^i)} \cdot \frac{\partial \hat{p}(x_b^1 | z_g^1)}{\partial z_g^1}}{\sum_{i=1}^K e^{f(z_g^1, z_d^i)} \cdot \left( \alpha \cdot \hat{p}(x_b^1 | z_g^1) + (1 - \alpha) \cdot \hat{p}(x_b^1 | z_d^i) \right)} \right.$$

$$\left. - \frac{\partial f(z_g^1, z_d^1)}{\partial z_g^1} + \frac{\sum_{i=1}^K e^{f(z_g^1, z_d^i)} \cdot \left( \alpha \cdot \hat{p}(x_b^1 | z_g^1) + (1 - \alpha) \cdot \hat{p}(x_b^1 | z_d^i) \right) \cdot \frac{\partial f(z_g^1, z_d^i)}{\partial z_g^1}}{\sum_{i=1}^K e^{f(z_g^1, z_d^i)} \cdot \left( \alpha \cdot \hat{p}(x_b^1 | z_g^1) + (1 - \alpha) \cdot \hat{p}(x_b^1 | z_d^i) \right)} \right],$$

$$\frac{\partial L_{\text{CLIP}}^g}{\partial z_d^1} = \mathbb{E}_{p(z_d^1, z_g^1, x_b^1)p(z_d^{2:K})} \left[ \frac{e^{f(z_g^1, z_d^1)} \cdot \frac{\partial \hat{p}(x_b^1 | z_d^1)}{\partial z_d^1}}{\sum_{i=1}^K e^{f(z_g^1, z_d^i)} \cdot \left( \alpha \cdot \hat{p}(x_b^1 | z_g^1) + (1 - \alpha) \cdot \hat{p}(x_b^1 | z_d^i) \right)} \right.$$

$$\left. - \frac{\partial f(z_g^1, z_d^1)}{\partial z_d^1} + \frac{e^{f(z_g^1, z_d^1)} \cdot \left( \alpha \cdot \hat{p}(x_b^1 | z_g^1) + (1 - \alpha) \cdot \hat{p}(x_b^1 | z_d^1) \right) \cdot \frac{\partial f(z_g^1, z_d^1)}{\partial z_d^1}}{\sum_{i=1}^K e^{f(z_g^1, z_d^i)} \cdot \left( \alpha \cdot \hat{p}(x_b^1 | z_g^1) + (1 - \alpha) \cdot \hat{p}(x_b^1 | z_d^i) \right)} \right],$$

$$\frac{\partial L_{\text{CLIP}}^g}{\partial z_d^i} = \mathbb{E}_{p(z_d^1, z_g^1, x_b^1)p(z_d^{2:K})} \left[ \frac{e^{f(z_g^1, z_d^i)} \cdot \frac{\partial \hat{p}(x_b^1 | z_d^i)}{\partial z_d^i}}{\sum_{i=1}^K e^{f(z_g^1, z_d^i)} \cdot \left( \alpha \cdot \hat{p}(x_b^1 | z_g^1) + (1 - \alpha) \cdot \hat{p}(x_b^1 | z_d^i) \right)} \right.$$

$$\left. + \frac{e^{f(z_g^1, z_d^i)} \cdot \left( \hat{p}(x_b^1 | z_g^1) + \hat{p}(x_b^1 | z_d^i) \right) \cdot \frac{\partial f(z_g^1, z_d^i)}{\partial z_d^i}}{\sum_{i=1}^K e^{f(z_g^1, z_d^i)} \cdot \left( \alpha \cdot \hat{p}(x_b^1 | z_g^1) + (1 - \alpha) \cdot \hat{p}(x_b^1 | z_d^i) \right)} \right].$$

Similar formulas can be derived for $\frac{\partial L_{\text{CLIP}}^t}{\partial z_d^1}$, $\frac{\partial L_{\text{CLIP}}^t}{\partial z_g^1}$, and $\frac{\partial L_{\text{CLIP}}^t}{\partial z_g^i}$.

The second line of each gradient formula contains $\partial f(z_g^1, z_d^i)/\partial z_g^1$ and $\partial f(z_g^1, z_d^i)/\partial z_d^i$ (and analogously $\partial f(z_d^1, z_g^i)/\partial z_d^1$ and $\partial f(z_d^1, z_g^i)/\partial z_g^i$). Thus it represents a standard component that optimizes the representations in a similar manner to CLIP, i.e. making the anchor and positive closer to each other and the anchor and negatives farther away in the latent space, but to different extents according to the reweighting factors in InfoCORE. The first line of each gradient formula contains $\partial \hat{p}(x_b^1|z_g^i)/\partial z_g^i$ and $\partial \hat{p}(x_b^1|z_d^i)/\partial z_d^i$. Thus it is a competing component that optimizes the representations so that their predictive power of the batch identifier becomes lower. Both components drive the representations towards a reduced batch effect.

This allows us to employ a hyperparameter $\lambda$ to regulate the extent to which each component contributes to the gradient update. We follow a similar practice as in the gradient reversal layer by Ganin & Lempitsky (2015). However, instead of reversing the gradient of $\hat{p}(x_b^1|z_g^i)$ w.r.t $z_g^i$ (and that of $\hat{p}(x_b^1|z_d^i)$ w.r.t $z_d^i$), we preserve the sign of the gradient while weighting the magnitude by $\lambda$.

During training, we update the encoders and the classifiers iteratively. To be more specific, in each iteration, we first update the parameters of the encoders (i.e., the drug structure encoder $\text{Enc}_d(.;\theta_d)$ and the drug screens encoder $\text{Enc}_g(.;\theta_g)$) to optimize the latent representations based on $L_{\text{CLIP}}$, while keeping the classifiers' parameters fixed. Then we fix the encoders and update the parameter of the batch classifiers $\hat{p}(x_b^1|z_g^1)$ and $\hat{p}(x_b^1|z_d^1)$ by optimizing the batch classification loss $L_{\text{CLF}}$.

## E  REVIEW OF CONTRASTIVE LEARNING AND THE INFONCE OBJECTIVE.

Contrastive learning (Tian et al., 2020; Oord et al., 2018; Bachman et al., 2019), as a self-supervised learning approach, aims to learn a representation space of high-dimensional data. Take the uni-modal contrastive learning of images as an example. Two views of the same image are generated to form a "positive pair" via data augmentations, one of which serves as the "anchor" and the other serves as the "positive". Meanwhile, to avoid representations collapsing to a single point, "negative" samples, which are views of different images, are included to form "negative pairs" with the anchor.

The success of contrastive learning has been connected to maximizing a lower bound of mutual information between the observation $X$ and the representation $Z$, which is lower bounded by $I(Z; Z^1)$ based on the data processing inequality, where $Z$ and $Z^1$ are latent representations of two views of the same observation $X$. Since mutual information is the KL divergence between the joint distribution and the product of marginal distributions, as proposed by Oord et al. (2018), we can maximize a lower bound of mutual information by optimizing the InfoNCE objective:

$$L_{\text{InfoNCE}} = \mathbb{E}_{p(z,z^1)p(z^{2:K})}\left[-\log \frac{e^{f(z,z^1)}}{\frac{1}{K}\sum_{i=1}^{K} e^{f(z,z^i)}}\right],$$

where $\{z^1\}_{i=2}^K$ are $K$ negative samples sampled i.i.d. from the marginal distribution $p(Z)$, i.e. latent representation of random images different from the anchor. Cosine similarity between $z$ and $z^i$ is usually used for the critic function $f$, projecting all the data observations into the representation space of a unit hypersphere.

In the case of bi-modal contrastive learning, the InfoNCE objective is generalized to two analogous terms, with one modality serving as the anchor and the other modality serving as the positive and the negatives. For instance, CLIP (Radford et al., 2021) learns representations of paired texts $T_1$ and images $I_1$, denoted as $Z_T^1$ and $Z_I^1$ using the following loss:

$$\frac{1}{2}\left[\mathbb{E}_{p(z_T^1,z_I^1)p(z_I^{2:K})}\left[-\log \frac{e^{f(z_T^1,z_I^1)}}{\frac{1}{K}\sum_{i=1}^{K} e^{f(z_T^1,z_I^i)}}\right] + \mathbb{E}_{p(z_T^1,z_I^1)p(z_T^{2:K})}\left[-\log \frac{e^{f(z_T^1,z_I^1)}}{\frac{1}{K}\sum_{i=1}^{K} e^{f(z_T^i,z_I^1)}}\right]\right]$$

In the presence of a sensitive attribute $X_b$, CCL by Ma et al. (2021) was proposed using the conditional contrastive learning objective tolower bound the conditional mutual information and reduce the information of the sensitive attribute from the representations by taking $X_b$ as the conditional variable. This leads to an objective resembling $L_{\text{InfoNCE}}$, but with expectation over the conditional distributions:

$$L_{\text{CCL}} = \mathbb{E}_{p(z,z^1)p(z^{2:K}|x_b)}\left[-\log \frac{e^{f(z,z^1)}}{\frac{1}{K}\sum_{i=1}^{K} e^{f(z,z^i)}}\right],$$

where $x_b$ is the value of the sensitive attribute corresponding to the anchor-positive pair $(z, z^1)$. In $L_{\text{CCL}}$, negative samples $\{z^i\}_{i=2}^K$ are sampled i.i.d. from the condition marginal distribution $p(Z|X_b = x_b)$. Similarly, this can be extended to the bi-modal case as follows:

$$\frac{1}{2}\left[\mathbb{E}_{p(z_T^1, z_I^1)p(z_I^{2:K}|x_b^1)}\left[-\log \frac{e^{f(z_T^1, z_I^1)}}{\frac{1}{K}\sum_{i=1}^K e^{f(z_T^1, z_I^i)}}\right] + \mathbb{E}_{p(z_T^1, z_I^1)p(z_T^{2:K}|x_b^1)}\left[-\log \frac{e^{f(z_T^1, z_I^1)}}{\frac{1}{K}\sum_{i=1}^K e^{f(z_T^i, z_I^1)}}\right]\right].$$

# F  ADAPTATION OF MULTIMODAL CONTRASTIVE LEARNING TO DRUG SCREENING DATA

Most multimodal contrastive learning methods are developed in the computer vision domain. We introduce several adaptions to enable application to drug screening data. We apply these modifications to all the benchmark models used in our experiments.

**Data Augmentation for Drug Screening.** Data augmentations often play a vital role in contrastive learning (Tian et al., 2020; Chen et al., 2020; He et al., 2020). However, data augmentation methods are missing for drug screening data with few replicates for each perturbation condition. Often, the experiment of applying a drug on a given cell line with a certain dosage and perturbation time is repeated only several times (typically 3-5 times). Based on the setup of drug screening experiments, we propose three kinds of data augmentations, which we find to effectively improve representation quality in practice:

1) *Adding Gaussian Noise.* Gaussian noise is added to $X_g$ before it is input into the encoder. The level of noise is controlled by a hyperparameter $\alpha_{\text{noise}}$. We set $\alpha_{\text{noise}} = 0.5$ for GE and $\alpha_{\text{noise}} = 0$ for CP in our experiments.

2) *Dirichlet Mixup.* Mixup (Zhang et al., 2017) generates a new data point by the weighted sum of existing samples. Given replicates of each experimental condition, we use mixup among the replicates; this helps filling in the support in high-dimensional data. To achieve broader coverage, we utilize Dir-mixup (Shu et al., 2021) to account for the multiple-replicates scenario. More precisely, we generated augmented samples according to the weighted average of the 3-5 replicates of each experimental condition (drug, cell line, dosage, etc.), and we samples the mixup weights $\mathbf{w}$ from a symmetric Dirichlet distribution with hyperparameter $\alpha_{\text{dir}}$: $\mathbf{w} \sim \text{Dirichlet}(\alpha_{\text{dir}})$. We set $\alpha_{\text{dir}} = 0.6$ for GE and $\alpha_{\text{dir}} = 0.8$ for CP in our experiments.

3) *Dropout (Masking).* Random dropout is conducted on the input data, which corresponds to masking expression level of certain genes, or values of certain features. Dropout proportions $\alpha_{\text{drop}}$ are set to 0.1 for both datasets.

**To Accommodate for Multiple Cell Lines.** In GE, each drug is screened on multiple cell lines, and drug effect varies across different cell lines. To learn a universal drug representation that gathers information from all cell lines, we make two modifications to existing methods (Jang et al., 2021) that directly use the average across cell lines or ignore the cell line labels.

Firstly, after applying a molecular structure encoder that takes molecular structure as input to generate drug embeddings, we utilize the cell line specific linear projection heads to map the universal embeddings into a context-aware latent space corresponding to the phenotype's cell line context. To be more precise, let $X_g^c$ denote the drug screening profile in cell line $c$ and $Z_g^c = \text{Enc}_g(X_g^c)$ the corresponding representation. Then the cell line contexualized drug representation $Z_d^c$ is calculated via $Z_d^c = \text{proj}_c(\text{Enc}_d(X_d))$ and both representations are optimized through the objective in Equation 4.

Secondly, since drug screening data is typically dominated by cell line effect, if we randomly sample data in the training batch, cell line related features will distract the model from learning drug-specific information. Therefore, we propose to sample training batches from experiments on the same cell line, so that the model can focus more on the difference in terms of drug structure. In practice, to achieve a good balance, we use training batches that iterate between a normal randomly sampled batch and a batch with all samples from the same cell line.

Additionally, we use the MoCo (He et al., 2020) framework for InfoCORE and CLIP and adapt it to our multimodal scenario with multiple cell lines. Two main adaptations are as follows: 1) maintaining the queue of momentum encoder outputs separately for each modality, and similarly the queue of batch distributions output by the momentum classifiers in InfoCORE; 2) constructing

cell line specific momentum queues for each of the nine cell lines, and extracting negative samples from cell line specific queues when the training batch has all samples from the same cell line (as discussed above). For CCL, in drug representation experiments, since the MoCo framework requires individual queues for each conditioning variable, i.e. each batch number, and the total batch number is large (97 for CP and about 1000 for GE), it is intractable in practice. We use the SimCLR (Chen et al., 2020) framework to overcome this. For better comparison, we also report the performance of CLIP and InfoCORE using the SimCLR framework in Appendix I. In the representation fairness experiments, since there are only 4 subgroups, we construct queues for each subgroup and use the MoCo framework directly.

**Ablation Study.** To showcase the effectiveness of these designs, we conducted an ablation study by removing each component from the CLIP model and calculating the retrieving accuracy in the GE dataset. The components include data augmentation (*aug*), the momentum contrastive learning framework (*MoCo*), the cell line specific projection heads (*cellproj*), and training batches iterating between random batch and cell line specific batch (*train bycell*). The results are shown in Table 5. For comparison, we also report the performance of the vanilla CLIP model with none of these adaptations. Note that all the results of CLIP and CCL reported in Table 2 and Table 3 are based on the modified model instead of the vanilla model, thereby providing a fair comparison reflecting the effectiveness of our InfoCORE algorithm to removing batch-related biases for multimodal molecular representation learning.

Table 5: Retrieving accuracy of different methods for drug screens based on gene expression data.

| Retrieval Library | *whole* | | | *batch* | | |
|---|---|---|---|---|---|---|
| Top N Acc (%) | N=1 | N=5 | N=10 | N=1 | N=5 | N=10 |
| InfoCORE | **6.48** | **19.13** | **27.53** | **14.16** | **33.93** | **47.15** |
| CLIP | 6.01 | 18.64 | 27.16 | 12.40 | 30.47 | 42.77 |
| CLIP (w/o *MoCo*) | 5.90 | 18.06 | 26.56 | 11.87 | 29.41 | 42.16 |
| CLIP (w/o *aug*) | 5.25 | 16.00 | 23.99 | 12.25 | 29.74 | 41.88 |
| CLIP (w/o *cellproj*) | 5.39 | 17.36 | 25.77 | 11.62 | 29.58 | 41.78 |
| CLIP (w/o *train bycell*) | 5.02 | 16.45 | 24.90 | 10.74 | 28.16 | 40.66 |
| vanilla CLIP | 4.64 | 14.86 | 22.16 | 11.32 | 28.46 | 40.55 |

## G  FAIRNESS CRITERIA

Following the approach by Ma et al. (2021), we use three different fairness criteria: equalized odds (EO), equality of opportunity (EOPP), and demographic parity (DP). Denote $l$ as the label and $\hat{l}$ as the model prediction, which are both binary variables. Denote $Z$ as the group identifier of sensitive attributes (also binary, considering the case of one binary value as the sensitive attribute). The definitions are provided in the following list:

1) Equalized odds (EO): EO calculates the sum of the difference (in absolute value) of the true positive rate and the false positive rate of the model predictions between two groups:

$$\text{EO} = |\mathbb{P}(\hat{l}=1|Z=0,l=1)-\mathbb{P}(\hat{l}=1|Z=1,l=1)|+|\mathbb{P}(\hat{l}=1|Z=0,l=0)-\mathbb{P}(\hat{l}=1|Z=1,l=0)|.$$

2) Equality of opportunity (EOPP): EOPP calculates the difference (in absolute value) of the true positive rate of the model prediction between the two groups:

$$\text{EOPP} = |\mathbb{P}(\hat{l}=1|Z=0,l=1) - \mathbb{P}(\hat{l}=1|Z=1,l=1)|.$$

3) Demographic parity (DP): DP calculates the difference (in absolute value) in model predictions between two groups:

$$\text{DP} = |\mathbb{P}(\hat{l}=1|Z=0) - \mathbb{P}(\hat{l}=1|Z=1)|.$$

These criteria can be calculated based on the above definitions for the case with a single binary sensitive attribute. We further adapt them to our setting where multiple subgroups exist and thus differences can be calculated for each pair of subgroups. Following the approach in Zhang et al. (2022), we calculate EO, EOPP, and DP for each subgroup pair, and then aggregate the pairwise values by taking the average. We observed in our experiments that using the maximum or median leads to similar results.

## H    DETAILS OF EXPERIMENTAL SETUP

**Simulation Study.** We generate the simulation input data according to the graphical model in Figure 2. To be more specific, we randomly assign a real effect identifier (1-5) and a batch effect identifier (1-25) to each of the 1250 samples. Each real effect category and batch effect category is represented by a 10-dimensional vector randomly sampled from the multivariate standard normal distribution. To introduce noise to the data, we also generate a 10-dimensional random Gaussian vector for each sample. Concatenating the three vectors into one 30-dimensional vector, and inputting it into two different random 2-layer Multi-Layer Perceptrons (MLP) corresponding to the two modalities, we obtain the simulated observations as outputs of the neural network. The simulated dataset is then split, with half utilized for training and the remaining half reserved for visualization in the 2-dimensional representation space. We use the same encoder architecture, i.e. a 3-layer MLP with hidden dimension size 128, for all the different models. For InfoCORE, the weighting hyperparameter $\alpha$ is set to be 0.09 and the gradient adjustment hyperparameter $\lambda$ is set to be 0.1.

**Representation Learning of Small Molecules.** We use L1000 gene expression profiles (Subramanian et al., 2017) (GE) and cell imaging profiles obtained from the Cell Painting assay (Bray et al., 2017) (CP) as pretraining datasets. In GE, 19,811 small molecules were screened across 77 cancer cell lines, and the drug effect was measured using L1000 profiles (i.e., the expression of 978 landmark genes was measured), with most data coming from 9 cell lines. We use the data from these nine cell lines. Since for most drugs only one perturbation dosage and perturbation time is available per cell line, for simplicity, we drop the few samples with multiple dosages or perturbation times. This results in 17,753 drugs and 82,914 drug-cell line pairs. We conduct standard batch correction as a preprocessing step by normalizing the gene expression vectors by the mean over the control groups in the same batch and then use this as model input. In CP, 30,204 small molecules are screened in one cell line (U2OS). We use the hand-crafted image features obtained by the popular CellProfiler method (McQuin et al., 2018), which gives rise to 701-dimensional tabular features after dropping missing values. The chemical structures are featurized using Mol2vec (Jaeger et al., 2018), which leads to 300-dimensional vector features. For both datasets and all the models, we use 3-layer MLPs as both the gene expression / cell imaging encoder and the molecular structure encoder, and we use a 256-dimensional representation embedding. For InfoCORE, we use 2-layer MLPs as the classifiers and set the hyperparameter $\alpha$ to be 0.33 for GE and 0.83 for CP, and $\lambda$ to be 0 for both datasets.

For the molecule-phenotype retrieval task, we calculate the representation embedding for each data sample in the validation set. Additionally, we calculate the drug structure representation of each molecule in the retrieval library (*whole* or *batch*, as discussed in Section 3.2). Then, for each data sample in the validation set, we rank the drugs in the drug library according to the cosine similarity between the drug structure embedding and the embedding of the data sample. Top N accuracy is then calculated based on the rank of the ground truth drug structure. In GE, the retrieval accuracy is calculated within each cell line, and we report the average over all cell lines. In CP, it is calculated on the U2OS cell line.

The retrieval library *whole* contains all molecules in the held-out set; so the accuracy among *whole* captures the model's general ability to pair drugs with their effects. The retrieval library *batch* contains held-out set molecules that have the same batch identifier as the retrieving target drug; so the accuracy among *batch* reflects the model's ability to distinguish drugs when the spurious features of the batch confounder are not available. Consider the extreme case: if a model only learns batch-related features, it can still have good accuracy in *whole* by randomly selecting drug candidates in the correct batch, but its performance in *batch* will be poor. Therefore, accuracies over both libraries as a whole reflect the model's ability of molecule-phenotype retrieval for drug discovery and drug repurposing.

In the downstream property prediction task, we follow the standard scaffold splitting procedure as suggested by Hu et al. (2020) for all classification tasks, and conduct scaffold splitting for the regression task of the PRISM dataset. For all pretrained models by different methods, we conduct a hyperparameter grid search for the learning rate and training epochs of the finetuning stage, select the ones with the best performance in the validation set (AUC for classification and R2 for regression), and report the mean and standard deviation on the test set over 3 random seeds.

**Representation Fairness.** For this, we use three fairness datasets with tabular features for binary classification: UCI Adult, Law School, and Compas. Race and gender features are binarized and

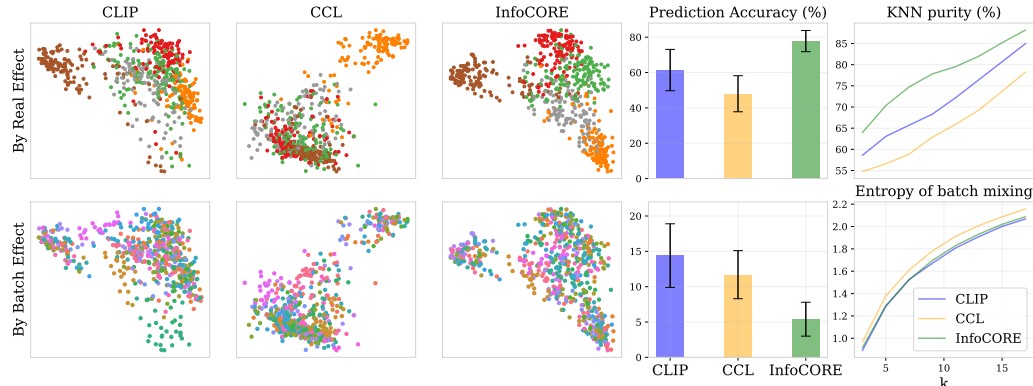

Figure 4: Visualization and quantitative metrics of the drug structure representations learned by different methods in the simulation experiment. The top row visualizations are colored by real effect category and the accuracy is calculated for the prediction of real effect category, while the bottom row uses batch identifier for both coloring and accuracy calculation. The farthest right column shows the KNN purity and entropy of batch mixing for different values of k.

combined to form the protected attributes, separating the data into four subgroups. Considering that minority groups often encounter data limitation issues, we subsample the training data to mimic an unbalanced population. To be specific, the training data consists of samples from one of the groups (white, male), (white, female), (black, male), (black, female) at the proportion of 10:2:2:1. At the pretraining stage, Gaussian noise is added as data augmentation for the tabular features. We use the MoCo framework for all three models. At the evaluation stage, we follow the data splitting procedure by Lahoti et al. (2020) and the approach taken by Ma et al. (2021) to freeze the encoder and train a small additional network (2-layer MLP) for label classification. Various fairness criteria are then measured based on the model predictions, with mean and standard deviation reported across 5 seeds.

## I    ADDITIONAL RESULTS.

**Simulation Study.** Analogously to Figure 3, the latent representations $Z_d$ for the molecular structure and the corresponding quantitative metrics are shown in Figure 4. Similar to Xu et al. (2021a), we calculated three quantitative metrics to evaluate the learned representations, including prediction accuracy, KNN purity, and entropy of batch mixing.

To calculate prediction accuracy, linear classifiers are trained to predict either the real effect category or the batch identifier based on the latent representation of drug screens and drug structures. Average accuracy and standard deviation are calculated based on 5-fold cross-validation. KNN purity calculates the average ratio of the intersection of the K-nearest neighbors based on the original data and the learned representation in each batch. It measures how well the representation retains the original structure of input data in each batch. Entropy of batch mixing calculates the average entropy of the empirical batch frequencies for the K-nearest neighbors of each drug. It measures whether drugs from different batches are well mixed in the representation space.

Table 6: Prediction accuracy (%) of the representation to real effect category and batch identifier.

| Representation | Drug Screens | | Drug Structure | |
|---|---|---|---|---|
| Target | Real Effect | Batch Identifier | Real Effect | Batch Identifier |
| CLIP | 44.2(7.0) | 8.6(2.6) | 61.4(11.7) | 14.4(4.5) |
| CCL | 45.1(4.6) | 7.8(1.9) | 48.0(10.2) | 11.7(3.4) |
| InfoCORE | **73.3(5.0)** | **5.6(3.6)** | **77.8(6.0)** | **5.4(2.4)** |

The same results apply for the simulated molecular structure representations. In the visualization, CLIP representations cannot fully mitigate batch effect, CCL representations fail to distinguish real

effect categories, while InfoCORE is able to recover the 5 real effect categories well. In terms of quantitative evaluation, InfoCORE outperforms CLIP and CCL in all metrics, especially in prediction accuracy of real effect category and KNN purity. The only exception occurs in the entropy of batch mixing for drug structure representation, where CCL has slightly higher entropy than CLIP and InfoCORE. However, the KNN purity of CCL is much worse. Considering the trade-off of these two metrics, CCL learns a better batch-independent representation, but fails to preserve the original data structure (i.e. real effect). The numerical results of prediction accuracy are shown in Table 6.

**Drug Representations.** As discussed in Appendix F, We use the MoCo (He et al., 2020) framework for CLIP and InfoCORE, and the SimCLR (Chen et al., 2020) framework for CCL (since MoCo requires individual queues for each conditioning variable and is not tractable for the drug screening datasets we experiment with). For better comparison with CCL using the SimCLR framework, we also run CLIP and InfoCORE with the SimCLR framework.

Table 7: Retrieving accuracy of different methods using either the MoCo or the SimCLR framework for gene expression and cell imaging screens and standard deviations over 3 random seeds.

| Dataset | Gene Expression (GE) | | | | | |
|---|---|---|---|---|---|---|
| Retrieval Library | *whole* | | | *batch* | | |
| Top N Acc (%) | N=1 | N=5 | N=10 | N=1 | N=5 | N=10 |
| Random | 0.03 | 0.13 | 0.27 | 1.58 | 7.90 | 15.81 |
| CLIP (MoCo) | 5.96(0.08) | 18.59(0.07) | 27.17(0.02) | 12.23(0.17) | 30.29(0.16) | 42.63(0.17) |
| CLIP (SimCLR) | 5.81(0.09) | 18.26(0.18) | 26.65(0.08) | 11.97(0.13) | 29.49(0.07) | 41.87(0.26) |
| CCL (SimCLR) | 1.93(0.10) | 5.85(0.18) | 8.37(0.04) | 12.76(0.29) | 32.39(0.16) | 45.77(0.08) |
| InfoCORE (MoCo) | **6.39(0.16)** | **18.99(0.19)** | **27.18(0.30)** | **14.03(0.33)** | **33.63(0.27)** | **46.78(0.38)** |
| InfoCORE (SimCLR) | 6.34(0.24) | 18.80(0.33) | 27.05(0.23) | 13.88(0.18) | 33.25(0.24) | 46.38(0.20) |

| Dataset | Cell Painting (CP) | | | | | |
|---|---|---|---|---|---|---|
| Retrieval Library | *whole* | | | *batch* | | |
| Top N Acc (%) | N=1 | N=5 | N=10 | N=1 | N=5 | N=10 |
| Random | 0.02 | 0.08 | 0.17 | 1.59 | 7.97 | 15.94 |
| CLIP (MoCo) | 7.23(0.05) | 20.95(0.28) | 28.89(0.34) | 13.20(0.10) | 37.78(0.36) | 52.72(0.16) |
| CLIP (SimCLR) | 6.99(0.16) | 21.09(0.24) | 29.16(0.29) | 12.96(0.28) | 37.58(0.59) | 52.87(0.31) |
| CCL (SimCLR) | 1.31(0.04) | 4.93(0.16) | 7.38(0.34) | 13.20(0.23) | 37.99(0.12) | 53.13(0.30) |
| InfoCORE (MoCo) | 6.93(0.26) | 20.65(0.25) | 28.22(0.14) | **13.26(0.10)** | **38.50(0.35)** | **53.13(0.08)** |
| InfoCORE (SimCLR) | **7.29(0.29)** | **21.15(0.33)** | **29.16(0.30)** | 13.16(0.12) | 38.24(0.36) | 53.00(0.25) |

Table 8: Performance of different methods using either the MoCo or the SimCLR framework on the molecular property prediction task.

| | Datasets | Classification (ROC-AUC %) ↑ | | | | | | | | Reg ($R^2$ %) ↑ |
|---|---|---|---|---|---|---|---|---|---|---|
| | | BBBP | BACE | ClinTox | Tox21 | ToxCast | SIDER | HIV | Avg. | PRISM |
| | # Molecules | 2039 | 1513 | 1478 | 7831 | 8575 | 1427 | 41127 | - | 3172 |
| | # Tasks | 1 | 1 | 2 | 12 | 617 | 27 | 1 | - | 5 |
| | Mol2vec | 70.7(0.4) | 82.9(0.7) | 84.9(0.3) | 76.0(0.1) | 74.4(0.5) | 64.9(0.3) | 77.7(0.1) | 75.9 | 8.5(0.7) |
| GE | CLIP (MoCo) | 73.5(0.4) | 86.1(0.4) | 89.6(2.1) | 77.3(0.0) | 75.7(0.6) | 63.7(0.6) | 77.7(0.6) | 77.6 | 13.9(0.4) |
| | CLIP (SimCLR) | 73.2(0.8) | 85.8(0.5) | 88.5(2.7) | 77.2(0.2) | 76.0(0.1) | 64.6(0.1) | 78.9(0.7) | 77.7 | 15.7(0.5) |
| | CCL (SimCLR) | 73.0(0.8) | 85.9(0.6) | 90.5(1.0) | 77.0(0.2) | 75.8(0.2) | 63.4(0.5) | 77.5(0.9) | 77.6 | **16.0(0.5)** |
| | InfoCORE (MoCo) | 73.5(0.3) | **86.6(0.3)** | **91.9(1.9)** | **77.4(0.4)** | 75.7(0.2) | 64.8(0.6) | 78.5(0.2) | 78.3 | 14.8(0.1) |
| | InfoCORE (SimCLR) | **73.6(0.3)** | 85.8(0.8) | 91.4(2.1) | 77.3(0.3) | **76.1(0.2)** | **65.3(0.6)** | **79.1(0.2)** | **78.4** | 14.9(0.1) |
| CP | CLIP (MoCo) | 73.4(0.8) | **85.2(0.4)** | 87.3(0.1) | 76.4(0.1) | 76.7(0.1) | 64.8(0.6) | 78.2(0.4) | 77.4 | 16.2(0.2) |
| | CLIP (SimCLR) | 74.0(0.6) | 84.2(0.4) | 88.8(0.4) | 77.0(0.2) | 76.7(0.2) | 64.6(0.8) | 79.1(0.4) | 77.8 | 15.6(0.2) |
| | CCL (SimCLR) | 73.7(0.5) | 84.9(0.9) | 87.7(1.8) | 75.9(0.3) | 75.7(0.4) | 65.2(0.4) | 79.3(0.3) | 77.5 | 14.7(0.3) |
| | InfoCORE (MoCo) | 74.0(0.8) | 85.0(0.2) | **89.3(0.5)** | 76.6(0.1) | 76.9(0.1) | 65.2(0.1) | 78.7(0.1) | 78.0 | 16.2(0.3) |
| | InfoCORE (SimCLR) | **74.3(0.5)** | 84.9(0.2) | 88.7(0.8) | **77.1(0.3)** | **77.0(0.1)** | **65.3(0.3)** | **80.0(0.8)** | 78.2 | **16.3(0.2)** |

We calculate the standard deviations of the retrieval accuracy for the molecule-phenotype retrieval task based on 3 random seeds. The results of retrieval accuracy for all models are reported in Table 7 and the results of property prediction are reported in Table 8.

**Representation Fairness.** Results of DP with different methods over 3 datasets are shown in Table 9. InfoCORE outperforms CLIP and CCL in terms of DP.

Table 9: Performance of various methods on representation fairness task, reported by DP (in percentage values).

| Method | UCI Adult | Law School | Compas |
|---|---|---|---|
| CLIP | 13.1(0.4) | 18.4(0.9) | 9.5(1.1) |
| CCL | 13.2(1.1) | 16.7(0.9) | 8.9(1.7) |
| InfoCORE | **12.4(0.5)** | **15.7(1.8)** | **8.2(0.8)** |

## J  VARIABLES DEFINITION

In this section, We provide a summary of the definitions of all variables used in our paper. Variables used in the single-sample setting in Section 2.2.1 are defined in Table 10. They are expanded to the multi-sample setting in Section 2.2.2, as defined in Table 11. The definition of anchor, positive, and negative samples in contrastive learning is reviewed in Appendix E.

Table 10: Definition of variables in the single-sample setting

| Variable | Definition |
|---|---|
| $X_d$ | $X_d \in \mathcal{D} \subseteq \mathbb{R}^{n_d}$ is the molecular structure of a drug. |
| $X_g$ | $X_g \in \mathcal{G} \subseteq \mathbb{R}^{n_g}$ is drug screens data reflecting the phenotypic change of cells induced by applying the drug (eg. gene expression and cell imaging). |
| $X_b$ | $X_b \in \mathcal{B}$ is a categorical variable representing the experimental batch identifier of the drug screening experiment. |
| $Z_d$ | $Z_d = \text{Enc}_d(X_d; \theta_d)$ is the representation of the drug structure; $\theta_d$ is the drug structure encoder parameter. |
| $Z_g$ | $Z_g = \text{Enc}_g(X_g; \theta_g)$ is the representation of the drug screens data; $\theta_g$ is the drug screen encoder parameter. |

Table 11: Definition of variables in the multi-sample setting

| Variable | Definition |
|---|---|
| $Z_d^1$ | The drug structure representation of the anchor-positive pair. |
| $Z_g^1$ | The drug screens representation of the anchor-positive pair. |
| $X_b^1$ | The experimental batch identifier of the anchor-positive pair. |
| $Z_d^i, i \in \{2:K\}$ | The drug structure representation of a negative sample. |
| $Z_g^i, i \in \{2:K\}$ | The drug screens representation of a negative sample. |

