# OpenReview forum: "Removing Biases from Molecular Representations via Information Maximization"
_ICLR.cc/2024/Conference — ICLR 2024 poster_

### Official Review · Reviewer_Tyeq · 2023-10-29

**Soundness:** 3 good
**Presentation:** 3 good
**Contribution:** 3 good
**Rating:** 6
**Confidence:** 2

**Summary:**

* A molecular representation learning method, InfoCORE, is introduced to mitigate the presence of confounders, such as batch effects, in multi-modal contrastive learning.
* Theoretical results were provided, showing that InfoCORE maximizes the variational lower bound on the conditional mutual information of the representations given the batch identifier.
* Empirical experiments were conducted to evaluate the proposed method along with baselines on one simulated dataset and two real drug screening datasets for two downstream tasks, including molecule-phenotype retrieval and molecular property prediction.
* A real-data experiment was presented to demonstrate that InfoCORE can be applied beyond drug discovery, improving fairness metrics when learning representations with sensitive attributes.

**Strengths:**

The proposed method is an extension of CLIP, an unconditional multi-model contrastive learning method, and CCL, a conditional contrastive learning method that does not address the confounder issue, thus tending to overcorrect for biological effects. The use of reweighting the negative samples differentially based on the posterior batch distribution solves the challenge of poor generalization in the case of limited negative samples for CCL. The results show a clear improvement in the metrics for both simulated data and real data tasks.

**Weaknesses:**

* It is not clear what is driving the algorithmic adjustments described in the Computational Considerations section, and how the choice of hyperparameters will impact the final results.
* In the simulation data experiment section, it can be relatively difficult to evaluate the quality of the learned representation by examining only the first two principal components since the variance in other dimensions can be concealed. Visualization through manifold learning techniques such as t-SNE or UMAP, or evaluation using various metrics, can provide more reliable insights.

**Questions:**

What is driving the algorithmic adjustment described in the computational considerations, what is the risk of doing so, and how the choice of the hyperparameter will impact the final results?

---

> ### Author Response · Authors · 2023-11-17
> **Response to Reviewer Tyeq**
>
> We appreciate the time and effort that the reviewer invested in reviewing our work. We are glad the reviewer valued the empirical evidence and technical innovation presented in our paper and recognized the key contributions of our work. Below, we offer detailed explanations in response to the reviewer’s questions.
>
> >What is driving the algorithmic adjustments described in the Computational Considerations section?
>
> The algorithmic adjustments are driven by the decomposition of the gradient of $L_{\text{CLIP}}$ with respect to the latent representations. By analyzing the gradient formulas, which we added in Appendix D, the gradient of $L_{\text{CLIP}}$ is driven by two components: one standard component containing $\frac{\partial f(z_g^1,z_d^i)}{\partial z_g^1}$ and $\frac{\partial f(z_g^1,z_d^i)}{\partial z_d^i}$ (and anagously $\frac{\partial f(z_d^1,z_g^i)}{\partial z_d^1}$ and $\frac{\partial f(z_d^1,z_g^i)}{\partial z_g^i}$) as denoted in the second line of each of the gradient formulas in Appendix D; and one competing component containing $\frac{\partial \hat{p}(x_b^1|z_g^i)}{\partial z_g^i}$ and $\frac{\partial \hat{p}(x_b^1|z_d^i)}{\partial z_d^i}$ as denoted in the first line of each of the formulas in Appendix D.
>
> The standard component optimizes the representations in a similar manner to CLIP, which makes the anchor and positive closer to each other and the anchor and negatives farther away in the latent space, but to different extents according to the reweighting factors in InfoCORE. The competing component optimizes the representations so that their predictive power of the batch identifier becomes lower. As demonstrated in Section 2.2.3, both gradient components drive the representations towards reduced batch effect. We use a hyperparameter $\lambda$ to regulate the extent to which each component contributes to the gradient update. In practice, since the competing component leads to a hard-to-optimize objective, we foud that using a smaller value of $\lambda$ generally works better. We added these explanations in Appendix D.
>
> To further understand the effect of $\lambda$ on performance, we report the retrieving accuracy in the CP dataset with varying $\lambda$ in the table below.
>
> |            |      | *whole* |       |       | *batch* |       |
> |------------|------|-------|-------|-------|-------|-------|
> | Top N Acc  | N=1  | N=5   | N=10  | N=1   | N=5   | N=10  |
> | $\lambda=0$   | 7.19 | 20.47 | 28.34 | 13.38 | 38.77 | 53.13 |
> | $\lambda=0.01$ | 7.29 | 19.97  | 27.41 | 13.28 | 38.00 | 52.66 |
> | $\lambda=1/4$ | 2.47 | 8.10  | 12.45 | 12.29 | 37.29 | 52.96 |
> | $\lambda=1/2$ | 1.44 | 4.93  | 7.34  | 11.90 | 34.87 | 50.07 |
> | $\lambda=3/4$ | 0.65 | 2.17  | 3.44  | 8.77  | 28.89 | 44.65 |
> | $\lambda=1$   | 0.60 | 2.36  | 3.57  | 8.62  | 28.57 | 44.83 |
>
> When $\lambda$ is large, the model performance drops due to the unstable competing objectives.
>
> >Visualization in the simulation experiment.
>
> We would like to clarify that in the simulation study the representations are actually 2-dimensional and no PCA is used. The first two principal components were only applied to the first column of Figure 3 since the original input is 10-dimensional. To avoid confusion, we removed the first column in Figure 3.
>
> In addition, as suggested by reviewer g7wR, we also added several quantitative metrics in Figure 3 and Figure 4, which complement our qualitative analysis. We added a detailed explanation of the metrics and results in Appendix I, and we provide the results of our analysis in the table below. The table shows the prediction accuracy (%) from the latent representation to predict either batch identifier or real effect category; InfoCORE clearly outperforms CLIP and CCL.
>
> |                |              | CLIP       | CCL        | InfoCORE       |
> |----------------|--------------|------------|------------|----------------|
> | Drug screens representation   | real effect  | 44.2(7.0) | 45.1(4.6) | **73.3(5.0)** |
> |  | batch id     | 8.6(2.6) | 7.8(1.9) | **5.6(3.6)** |
> | Drug structure representation | real effect  | 61.4(11.7) | 48.0(10.2) | **77.8(6.0)** |
> |  | batch id     | 14.4(4.5) | 11.7(3.4) | **5.4(2.4)** |

---

> > ### Comment · Reviewer_Tyeq · 2023-11-20
> > **Thank you for the reply**
> >
> > Thank the authors for replying to all my questions. My concerns are all addressed and my score will remain unchanged.

---

### Official Review · Reviewer_FJph · 2023-10-29

**Soundness:** 3 good
**Presentation:** 4 excellent
**Contribution:** 3 good
**Rating:** 8
**Confidence:** 3

**Summary:**

The paper presents a way to remove batch effect in contrastive learning of multimodal data while drawing negative samples from its marginal distribution. The weighting adjustment for negative samples is semi-parametric and it ensures that the weighting for unlikely negative samples (w.r.t. a batch number) does not shrink to zero. The model is primarily established in the context of molecular representation learning, but could be generalized to the removal of sensitive information in representation fairness problems. In experiments, the proposed model is examined in both context and overall achieves a better performance compared to the benchmarks.

**Strengths:**

Paper is well written: review of prior work is given and comparison to prior work is clear, walkthrough of derivation is intuitive and easy to follow. Figures of the graphical model and computational workflow are presented in a clean fashion that aids understanding the framework. The method is adequately novel and addressed the negative sample supply problem in conditional contrastive learning: the framework is capable of conducting training with batch data randomly sampled across different experiments. Experiments are done thorough.

**Weaknesses:**

1. The computational considerations for $L_{CLF}$ should be illustrated. $L_{CLF}$ by itself is not tractable without the true sampling distribution $p(x_b | z_d)$ and $p(x_b | z_g)$, and is not exactly a classification loss. Without any clear statement that you're replacing $L_{CLF}$ with conditional density learning through classification loss, there isn't any direct justification for factoring $ \mathbb E_{p(z_d^1, z_g^1, x_b^1)} \log \frac{(\hat{p}(x_b^1 | z_d^1, z_g^1))^2}{p(x_b | z_d^1)p(x_b | z_g^1)}$ into $C - L_{CLF}$.

2. Line 2 and line 3 of Table 1 should be merged, or line 3 should be just removed. Reweighting of samples is an adjustment you have to make for large supply of negative samples to work, it is not a standalone contribution or an advantage. The current formulation of the table is slightly misleading.

3. Figure 1. does not demonstrate the classification workflow clearly. The figure and the yellow coloring on the right grid makes it seem like you're optimizing $M_{ii}$ w.r.t. batch identifiers.

**Questions:**

See weaknesses.

---

> ### Author Response · Authors · 2023-11-17
> **Response to Reviewer FJph**
>
> We thank the reviewer for their valuable feedback. We are pleased that the reviewer appreciated the presentation and technical novelty of our paper and recognized the key contributions of our paper.  In response to the reviewer’s comments, we updated the manuscript providing the following clarifications:
>
> >1. Computational considerations for $L_{\text{CLF}}$.
>
> >$L_{\text{CLF}}$ by itself is not tractable and is not exactly a classification loss
>
> Since $D_\text{KL} \left (p\left(x_b^1|z_d^1\right) \Vert \hat{p}\left(x_b^1|z_d^1\right) \right )=H(p(x_b^1|z_d^1))+CE(p(x_b^1|z_d^1), \hat{p}(x_b^1|z_d^1))$, $L_{\text{CLF}}$ can be factorized into the cross entropy loss of the classifiers $\hat{p}(x_b^1|z_d^1)$ and $\hat{p}(x_b^1|z_g^1)$, and an additional constant.
>
> >Justification for factorizing $\mathbb{E}_{p(z_d^1,z_g^1,x_b^1)} \log \frac{(\hat{p}(x_b^1|z_d^1,z_g^1))^2}{p(x_b|z_d^1)p(x_b|z_g^1)}$ into $C-L\_{\text{CLF}}$.
>
> Note that using the following transformation, $\mathbb{E}_{p(z_d^1,z_g^1,x_b^1)} \log \frac{(\hat{p}(x_b^1|z_d^1,z_g^1))^2}{p(x_b|z_d^1)p(x_b|z_g^1)}$ can be factorized into $C-L\_{\text{CLF}}$:
>
> $\mathbb{E}_{p(z_d^1,z_g^1,x_b^1)} \log \frac{(\hat{p}(x_b^1|z_d^1,z_g^1))^2}{p(x_b|z_d^1)p(x_b|z_g^1)} = \mathbb{E}\_{p(z_d^1,z_g^1,x_b^1)} \log \frac{(\hat{p}(x_b^1|z_d^1,z_g^1))^2}{\hat{p}(x_b|z_d^1)\hat{p}(x_b|z_g^1)} \cdot \frac{\hat{p}(x_b|z_d^1)}{p(x_b|z_d^1)} \cdot \frac{\hat{p}(x_b|z_g^1)}{p(x_b|z_g^1)}$
>
> $= \mathbb{E}\_{p(z_d^1,z_g^1,x_b^1)} \log \frac{(\hat{p}(x_b^1|z_d^1,z_g^1))^2}{\hat{p}(x_b|z_d^1)\hat{p}(x_b|z_g^1)} - \mathbb{E}\_{p(z_d^1,x_b^1)} \log \frac{p(x_b|z_d^1)}{\hat{p}(x_b|z_d^1)} - \mathbb{E}\_{p(z_g^1,x_b^1)} \log \frac{p(x_b|z_g^1)}{\hat{p}(x_b|z_g^1)}$
>
> The last two terms are the expectation of KL-divergence between the posterior probability and the estimated one:
> $\mathbb{E}\_{p(z_d^1,x_b^1)} \left [ \log \frac{p(x_b^1|z_d^1)}{\hat{p}(x_b^1|z_d^1)}\right] = \mathbb{E}\_{p(z_d^1)} \left [ \mathbb{E}\_{p(x_b^1|z_d^1)} \left [ \log \frac{p(x_b^1|z_d^1)}{\hat{p}(x_b^1|z_d^1)}\right]\right] = \mathbb{E}\_{p(z_d^1)} \left [ \displaystyle D\_\text{KL}\left (p\left(x_b^1|z_d^1\right) \Vert \hat{p}\left(x_b^1|z_d^1\right) \right ) \right]$
>
> $\mathbb{E}\_{p(z_g^1,x_b^1)} \left [ \log \frac{p(x_b^1|z_g^1)}{\hat{p}(x_b^1|z_g^1)}\right] = \mathbb{E}\_{p(z_g^1)} \left [ \mathbb{E}\_{p(x_b^1|z_g^1)} \left [ \log \frac{p(x_b^1|z_g^1)}{\hat{p}(x_b^1|z_g^1)}\right]\right] = \mathbb{E}\_{p(z_g^1)} \left [ \displaystyle D\_\text{KL}\left (p\left(x_b^1|z_g^1\right) \Vert \hat{p}\left(x_b^1|z_g^1\right) \right ) \right]$
>
> Thus, $\mathbb{E}_{p(z_d^1,z_g^1,x_b^1)} \log \frac{(\hat{p}(x_b^1|z_d^1,z_g^1))^2}{p(x_b|z_d^1)p(x_b|z_g^1)} = 2\cdot(C-L\_{\text{CLF}})$.
>
> To clarify these points, we updated the detailed derivations in Appendix B accordingly.
>
> >2. Line 2 and line 3 of Table 1 should be merged.
>
> We would like to clarify that Line 3 (Per-sample weighting) is not exactly the same as Line 2 (Large Supply of Negatives) since even if we only consider the samples from the same batch as in CCL, InfoCORE will weight each sample differently but CCL will weight them equally.
>
> >3. Figure 1 does not demonstrate the classification workflow clearly.
>
> We thank the reviewer for this comment. To clarify the figure, we removed the yellow coloring on the right grid in Figure 1.

---

> ### Comment · Reviewer_FJph · 2023-11-20
> **Response to author**
>
> Thanks for the response. Let me clarify.
>
> 1. I just realized that I misunderstood the setup and $p(z_g, z_d, x_b)$ is actually the distribution over constructed latents rather than true latens. In this case I agree it is tractable. I was saying justification as in the reasoning for the factorization, not technical justification. But again, such justification is not required if $L_{CLF}$ is tractable.
>
> 2. Your clarification is correct but does not address my concern. The weighting is different but that's a modification you made as a direct result of contribution 2, i.e. in order for contribution 2. to work, you must have reweighting. I do not agree that it is a standalone contribution just because it is different. What is the advantage of having an estimated weight instead of equal weight if you only consider samples from the same batch?

---

> ### Author Response · Authors · 2023-11-20
> **Response to the Latest Comment by Reviewer FJph**
>
> We thank the reviewer for carefully reading and helping us improve our manuscript.
>
> Regarding point 1, $p(z_g, z_d, x_b)$ is indeed the distribution over the constructed latents and thus $L_{\text{CLF}}$ is tractable. We added a sentence in the manuscript to clarify this further.
>
> Regarding point 2, we understand the reviewer’s perspective and decided to remove line 3 from Table 1 in the updated manuscript.

---

> > ### Comment · Reviewer_FJph · 2023-11-20
> > **Response to author**
> >
> > Thanks for the response. All my concerns are addressed and my score remain unchanged.

---

### Official Review · Reviewer_JEFo · 2023-10-31

**Soundness:** 2 fair
**Presentation:** 2 fair
**Contribution:** 2 fair
**Rating:** 6
**Confidence:** 3

**Summary:**

This paper studies multimodal molecular representation learning. The authors analyze the confounding batch effects to the learned molecular representations, and adopt a conditional mutual information maximization objective among the modalities, called the Information maximization approach for COnfounder Removal (InfoCORE). InfoCORE also reweighs the negative samples in the InfoNCE objective differentially based on the posterior batch distribution, to resolve the issue of insufficient negatives.

**Strengths:**

(+) The studied problem is critical;

(+) The proposed method is interesting;

**Weaknesses:**

(-) The paper is hard to follow;

(-) The novelty is limited;

(-) The improvements are limited;

(-) Some related works have not been compared or discussed;

**Questions:**

1. The paper is hard to follow:
- **All of the variables** in the paper have not been defined;
- The problem is undefined;
- The bias issue caused by the batching effects seems to be interesting, but does it really exist? Are there any realistic examples for the biases?
- What does it mean by claiming X_b as a irrelevant attribute?
- Why does equation (1) “emphasizes the shared features of the two modalities that are unrelated to batch” and “thus emphasizing the bioactivity of a drug”?
- What is the exact objective of InfoCORE?
- What is the algorithm of InfoCORE?
- The information and the caption of Figure 3 is hard to read. It seems the batching identifier has not influence to the learned representations for all methods.

2. The novelty is limited. It seems the main objective InfoCORE comes from CCL. What are the differences between InfoCORE and CCL?

3. The improvements are limited.
- In Table 2, the improvements of InfoCORE over CLIP or CCL is marginal while no standard deviations are given. Even the vanilla objective CLIP can outperform InfoCORE.
- In Table 3, InfoCORE has little-to-no improvements over baselines, when considering the standard deviations.

4. Some related works including debiasing or invariant molecular representation learning such as [1,2,3,4,5] have not been compared or discussed.


**References**

[1] Discovering invariant rationales for graph neural networks, ICLR 2022.

[2] Learning causally invariant representations for out-of-distribution generalization on graphs, NeurIPS 2022.

[3] Learning substructure invariance for out-of-distribution molecular representations, NeurIPS 2022.

[4] Interpretable and Generalizable Graph Learning via Stochastic Attention Mechanism, ICML 2022.

[5] Debiasing Graph Neural Networks via Learning Disentangled Causal Substructure, NeurIPS 2022.

---

> ### Author Response · Authors · 2023-11-17
> **Response to Reviewer JEFo**
>
> We thank the reviewer for taking the time to evaluate our manuscript and for the comments. We have carefully considered the reviewer’s comments and made revisions to clarify and improve the paper, especially regarding the clarity and exposition of our key arguments and methodologies. In summary, our paper focuses on learning molecular representations from multiple information sources, specifically drug screening data. InfoCORE is introduced to improve representations by explicitly countering confounding batch effects from non-biological sources. Such effects are typical also in other biological experimental data. This is a very different problem from supervised graph classification illustrated in [1,2,3,4,5] that the reviewer referred to. We provide point-by-point explanations to all of the reviewer’s comments below.
>
> >**1. The paper is hard to follow:**
>
> >Definition of the variables.
>
> The variables $X_d, X_g, Z_d, Z_g, X_b$ are defined in Section 2.1. We added the definition of $Z_d^i, Z_g^i, X_b^i, \ i \in 1,...,K$ in Section 2.2.2. Variables with superscript $i, \ i \in 1,...,K$ are the multi-sample analog of those in the single-sample case. Similar to the definition in the multi-sample InfoNCE objective in contrastive learning, $i=1$ denotes the variables associated with the anchor-positive pair, and $i>1$ denotes the variables for the negative samples (drugs other than the one in the anchor-positive pair).
>
> To further clarify the definition of all variables used in our paper, we added a lookup table in Appendix J.
>
> >Definition of the problem.
>
> The problem is defined in detail in the introduction as well as in Section 2.1). Briefly, the overall goal of our paper is to learn better molecular representations from different information sources. These representations can then be applied to various downstream tasks in drug discovery related prediction tasks. More precisely, we use the data from drug screening experiments to learn bioactivity related representations of drugs. Since batch effects dominate many biological experimental data, including drug screens, batch-related spurious features severely bias the learned representations. We translate the goal of learning de-biased representations into a conditional mutual information objective, and propose a lower bound criterion as an effective training objective.
>
> >Training objective and the algorithm of InfoCORE.
>
> - The exact objective of InfoCORE is described in Equation 4, i.e. $L_{\text{InfoCORE}} = L_{\text{CLIP}} + L_{\text{CLF}}$, where $L_{\text{CLIP}}$ and $L_{\text{CLF}}$ are defined in Proposition 2. $L_{\text{CLIP}}$ resembles the symmetrical InfoNCE loss in CLIP, but with negative samples reweighted according to the posterior batch distribution estimated by the two batch classifiers ($\bar{D}$ and $\bar{G}$ in Figure 1). $L_{\text{CLF}}$ is the classification loss for training the classifiers to predict the batch identifier.
>
> - The algorithm of InfoCORE is described in Section 2.2.2 and Section 2.2.3 and illustrated in Figure 1. Briefly, during training, the encoders and classifiers are updated iteratively. In each iteration, we first update the encoder parameters (i.e. the drug structure encoder
> $\text{Enc}\_{d}(.;\theta_d)$ and the drug screen encoder $\text{Enc}\_g(.;\theta_g)$) to optimize the latent representations using the new $L_{\text{CLIP}}$ objective, while keeping the classifier parameters fixed. Then we fix the encoders and update the parameters of the auxiliary batch classifiers $\hat{p}(x_b^1|z_g^1)$ and $\hat{p}(x_b^1|z_d^1)$ via optimizing the batch classification loss $L_{\text{CLF}}$.
>
> Based on the reviewer's comments, we updated the introduction and methods section in the revised manuscript to improve its readability and clarity. We welcome additional comments to further improve the clarity.

---

> ### Author Response · Authors · 2023-11-17
> **Response to Reviewer JEFo (continued)**
>
> >Realistic examples for bias issue caused by the batching effects. What does it mean by claiming X_b as an irrelevant attribute?
>
> Batch effects are common and challenging in biological experimental data. Substantial research has gone into mitigating such effects (e.g., [6,7,8,9,10]). Please also refer to the second paragraph in the related work section. Broadly speaking, batch effect is a change in the data that is caused by non-biological factors in an experiment [6], such as temperature, humidity, the person conducting the experiment, etc. Thus this effect is irrelevant to the real biological effect of a drug that we hope to identify in the representations $Z_d$ and $Z_g$.
>
> To better explain this, we added the following sentence in the introduction: “Batch effects refer to non-biological effects introduced into the data through the experimental measurement process.” Specifically, in the context of drug screening, each “batch” refers to a (384-well) plate used in the experiments (384 drugs are tested at once in one plate; experiments on different plates are performed e.g. by different people, on different days, etc. and thus show non-biological variation, termed “batch effect”; however, complicating the situation is the fact that variation between plates is not always non-biological as we discussed in Section 2.1; this is due to the fact that similar drugs are often tested on the same plate and different cell lines are tested in different plates).
>
> [6] Adjusting batch effects in microarray expression data using empirical Bayes methods. Biostatistics, 2007.
>
> [7] Fast, sensitive and accurate integration of single-cell data with harmony. Nature Methods, 2019.
>
> [8] Efficient integration of heterogeneous single-cell transcriptomes using scanorama. Nature Biotechnology, 2019.
>
> [9] Batch effects in single-cell rna-sequencing data are corrected by matching mutual nearest neighbors. Nature Biotechnology, 2018.
>
> [10] Deep learning enables accurate clustering with batch effect removal in single-cell rna-seq analysis. Nature Communications, 2020.
>
> > Why does equation (1) “emphasizes the shared features of the two modalities that are unrelated to batch” and “thus emphasizing the bioactivity of a drug”?
>
> When maximizing the conditional mutual information objective, we are optimizing the representations $Z_d$ and $Z_g$ so that their batch-irrelevant shared information is maximized. Since we condition on $X_b$, we disallow any features tied to $X_b$ to contribute to the mutual information we are maximizing. Moreover, since the representations are kept in a fixed dimension, there is an explicit bias against incorporating any irrelevant attributes that do not contribute to mutual information.
>
> $X_g$ corresponds to drug screening data (for example, the gene expression change of cells after applying a drug) and thus reflects the bioactivity of a drug (i.e. the effect of a drug in the cellular context). Thus, after removing non-biological batch effects $X_b$, the shared information captures the bioactivity of a drug.
>
> > The information and the caption of Figure 3 is hard to read. It seems the batching identifier has no influence on the learned representations for all methods.
>
> Detailed descriptions of the simulation setup are given in the main text of Section 3.1. We updated the caption of Figure 3 and Section 3.1 to better explain the setting and results as detailed in response to the comments below.
>
> - *On the Simulation setup*
>
> In a nutshell, the scatter plots show the 2-dimensional latent representations of the simulated input data learned by each method. Each data point is labeled with one of five real effect categories and one of 25 batch identifiers. Top row figures color dots by real effect category, and bottom row figures by batch identifier. Therefore, an ideal representation should separate real effect categories well but not be impacted by the batch identifier.

---

> ### Author Response · Authors · 2023-11-17
> **Response to Reviewer JEFo (continued 2)**
>
> - *Evidence of influence of batch identifier on the representation*
>
> The representation learned by CLIP is distorted by the batch effects (for example, the green batch cluster in the top right corner). The batch identifier's impact is also evident in the new prediction accuracy metric below.
>
> As suggested by reviewer g7wR, we added several quantitative metrics in Figure 3 and Figure 4, with a detailed explanation of the metrics and results depicted in Appendix I. The prediction accuracy (%) from the latent representation to predict either batch identifier or real effect category is shown in the following table. InfoCORE has a clear advantage compared to CLIP and CCL, and the prediction accuracies to the batch identifier for CLIP and CCL are significantly larger than random guessing (4%), showing that they are influenced by the batch identifier.
>
> |                |              | CLIP       | CCL        | InfoCORE       |
> |----------------|--------------|------------|------------|----------------|
> | Drug screens representation   | real effect  | 44.2(7.0) | 45.1(4.6) | **73.3(5.0)** |
> |  | batch id     | 8.6(2.6) | 7.8(1.9) | **5.6(3.6)** |
> | Drug structure representation | real effect  | 61.4(11.7) | 48.0(10.2) | **77.8(6.0)** |
> |  | batch id     | 14.4(4.5) | 11.7(3.4) | **5.4(2.4)** |
>
> >**2. The novelty is limited.** Differences between InfoCORE and CCL.
>
> We respectfully disagree with the reviewer. Although both InfoCORE and CCL aim to maximize the conditional mutual information objective, computationally tractable estimation of conditional mutual information is an area that has been extensively researched and is known to be quite hard [1,2,3]. InfoCORE and CCL are based on **very different training objectives**.
>
> * CCL uses the InfoNCE loss with negative samples drawn from the conditional distribution to approximate the conditional mutual information:
> $$L_{\text{CCL}} = \frac{1}{2} \left[ \mathbb{E}\_{p(z_d^1,z_g^1) p(z_d^{2:K}|x_b^1)}\left[- \log \frac{e^{f(z_g^1, z_d^1)}}{\frac{1}{K}\sum\_{i=1}^K e^{f(z_g^1, z_d^i)}}\right] + \mathbb{E}\_{p(z_d^1,z_g^1) p(z_g^{2:K}|x_b^1)}\left[-\log \frac{e^{f(z_g^1, z_d^1)}}{\frac{1}{K}\sum\_{i=1}^K e^{f(z_g^i, z_d^1)}} \right]\right]$$
> However, it does not work well when there are limited observations for each condition (which is the case in our biological experiments, where there are only about 300 drugs in each batch and there are hundreds of batches in total).
>
> * Instead, InfoCORE establishes a different lower bound for conditional mutual information, with $L_{\text{InfoCORE}} = L_{\text{CLIP}}+L_{\text{CLF}}$ as the training objective:
> $$
> L_{\text{CLIP}} = \frac{1}{2}\left[ \mathbb{E}\_{p(z_d^1,z_g^1,x_b^1)p(z_d^{2:K})} \left [ -\log \frac{e^{f(z_g^1,z_d^1)}}{\frac{1}{K} \sum\_{i=1}^K e^{f(z_g^1,z_d^i)} \cdot \hat{p}(x_b^1|z_g^1,z_d^i)} \right ] \\ + \mathbb{E}\_{p(z_d^1,z_g^1,x_b^1)p(z_g^{2:K})} \left [ -\log \frac{e^{f(z_g^1,z_d^1)}}{\frac{1}{K} \sum\_{i=1}^K e^{f(z_g^i,z_d^1)} \cdot \hat{p}(x_b^1|z_g^i,z_d^1)} \right ] \right]
> $$
> $$L_{\text{CLF}} = \frac{1}{2} \left[ \mathbb{E}\_{p(z_d^1)} \left [ D\_\text{KL} \left (p\left(x_b^1|z_d^1\right) \Vert \hat{p}\left(x_b^1|z_d^1\right) \right ) \right] + \mathbb{E}\_{p(z_g^1)} \left [ D\_\text{KL} \left( p\left(x_b^1|z_g^1\right) \Vert \hat{p}\left(x_b^1|z_g^1\right) \right) \right] \right]$$
> This objective is based on information theoretic derivation and can make use of all the negative samples by adaptively reweighting them according to reweighting factors $\hat{p}(x_b^1|z_g^1,z_d^i)$ and $\hat{p}(x_b^1|z_g^i,z_d^1)$.
>
> [1] Mutual information neural estimation. ICML 2018.
>
> [2] Understanding the limitations of variational mutual information estimators. ICLR 2020.
>
> [3] High-order conditional mutual information maximization for dealing with high-order dependencies in feature selection. Pattern Recognition 2022.
>
> >**3. Performance.**
>
> >- In Table 2, the improvements of InfoCORE are marginal while no standard deviations are given.
>
> In Table 2, we calculate retrieving accuracy over two retrieval libraries, "*whole*" and "*batch*". It is important to note that accuracies **over both libraries as a whole** reflect the model’s ability of molecule-phenotype retrieval for drug discovery and drug repurposing. InfoCORE is **the only** model that has strong performance **in both libraries**. In comparison, CCL has very poor performance in "*whole*", and CLIP has significantly worse performance in "*batch*" (for example, InfoCORE has 15% gain in top 1 accuracy and 11% gain in top 5 accuracy over batch in the GE dataset compared with CLIP). We also highlighted this in the revised manuscript.
>
> We also added the standard deviation of the retrieving accuracies based on 3 random seeds in Table 6 in Appendix I.

---

> ### Author Response · Authors · 2023-11-17
> **Response to Reviewer JEFo (continued 3)**
>
> >- In Table 3, InfoCORE has little-to-no improvements over baselines.
>
> It is known that the property prediction task in Table 3 is hard and even small gains on these benchmark datasets are significant. For example in reference [1], the proposed model's gains over the baselines are more marginal than in our paper, and in [2][3], the proposed model does not show consistent gains in most datasets.
>
> [1] Uni-Mol: A Universal 3D Molecular Representation Learning Framework. ICLR 2023.
>
> [2] Graph contrastive learning with augmentations. NeurIPS 2020.
>
> [3] Let Invariant Rationale Discovery Inspire Graph Contrastive Learning. ICML 2022.
>
>
> >**4. Related works.**
> >Some related works including debiasing or invariant molecular representation learning such as [1,2,3,4,5] have not been compared or discussed.
>
> These related works focus on learning _invariant_ or interpretable features in the _supervised_ graph classification task. However, as discussed in our problem setting, InfoCORE learns unbiased molecular representations in an _unsupervised setting_, where the only supervision comes from another paired high-dimensional data modality and is not explicitly related to the targets in the downstream tasks. We added a brief explanation contrasting InfoCORE and these related works in Section 4 summarizing the following discussion:
>
> In [1-5], the label $Y$ is available during training and is the same as the label targeted in OOD generalization, while in InfoCORE, labels in the downstream tasks are not available during pretraining. Besides, [1-5] focus on separating the spurious features and invariant features of a graph structure (eg. certain subgraphs), while the spurious features in InfoCORE are caused by the batch effect in biological experiments, which is unrelated to graph structure.
>
> Although it is conceivable that the information objective in [2,3,4] could be adapted to our setting by replacing $Y$ with $X_g$, the adaption would be highly non-trivial. This is because $Y$ is the classification label while $X_g$ is a high-dimensional experimental data vector, and the goal of InfoCORE as a pretrained model is different from [2,3,4]. Additionally, one of the main contributions of InfoCORE is the tractable lower bound of conditional mutual information as the training objective.
>
> In the following, we provide a detailed comparison of InfoCORE to each of the referenced works.
>
> [1] uses a distribution intervener on a graph to generate multiple interventional distributions. However, it is not applicable to our case since generating an interventional or counterfactual phenotype of a drug in a new batch is non-trivial and very hard to evaluate.
>
> [2] optimizes the graph features to be independent with $E$. However, as shown in our graphical model in Figure 2, $X_b$ is a confounder between $X_d$ and $X_g$, thus the optimal features could still be correlated with $X_b$.
>
> [3] optimizes a similar objective as [2], but constrains the conditional independence between $Y$ and $E$ conditioned on the learned graph features. However, in our graphical model, there exists a direct dependence between both modalities and $X_b$, and the correlation between $X_g$ and $X_b$ via non-biological batch effect remains even after taking the real biological effect into account.
>
> [4] focuses on the interpretability of graph classifiers, which is a different problem. It selects the substructure of an input graph that is most relevant to the label $Y$, which minimizes the mutual information between the graph feature and the original graph. However, in our problem, we hope the representations maintain as much shared information between the two modalities as possible, as long as they are not through the backdoor path via $X_b$.
>
> [5] considers a supervised learning setting similar to [2][3], and the algorithm is specifically designed for a graph as input data. Similar to [1], it requires counterfactual unbiased sample generation which is not applicable in our case. Also, the spurious correlation that we want to minimize in the representation is explicitly defined as the batch effect $X_b$, while that in [5] are implicit subgraphs and generally unrelated to batch effect.

---

> ### Comment · Reviewer_JEFo · 2023-11-20
> **Thank you for the reply**
>
> Thank the authors for the detailed explanations, which resolve most of my concerns. Nevertheless, I still feel that the system of notations can be better defined. The novelty remains limited, given the limited performance gains. Therefore, I have adjusted my rating accordingly.

---

> ### Author Response · Authors · 2023-11-20
> **Response to the Latest Comment by Reviewer JEFo**
>
> We thank the reviewer for their reassessment and acknowledgement of our detailed explanations, along with increasing their score. We are pleased to hear that these clarifications have addressed most of their concerns. We appreciate the suggestion about further improving our notation and will add more additional clarifications in the revised manuscript.

---

### Official Review · Reviewer_g7wR · 2023-11-01

**Soundness:** 3 good
**Presentation:** 3 good
**Contribution:** 2 fair
**Rating:** 6
**Confidence:** 3

**Summary:**

The authors tackle the problem of removing batch effect in high-throughput drug screening.
The proposed method InfoCORE leverages a tractable information theoretic framework by establishing a variational lower bound on the conditional mutual information of some latent representation (obtained by a NN) given a "batch identifier" (i.e. the batch effect).
The method is rather generic in the sense that it can handle multiple modalities such as gene expression levels, imaging profiles, along with the chemical structure of the drug at hand.
Unlike many methods that try to learn a generic representation, as a pre-processing task, InfoCORE trains end-to-end for a final task (e.g. molecular property prediction, molecule-phenotype retrieval).
While very closely related to existing frameworks such as CLIP and CCL, InfoCORE's main contribution is to adaptively reweigh negative samples (in a contrastive learning sense) in the objective, so as to make better use of low sample sizes for rare conditions.
While adaptively adjusting the sample weights, the resulting approach is proven to be optimizing a lower bound for the conditional mutual information.
Last, the method is benchmarked against its very related methods CLIP and CCL on simulated data, representation learning of small molecules and on standard UCI datasets for fairness of the learnt representation.

**Strengths:**

While building on already very established work, InfoCORE is addressing an important shortcoming of InfoNCE and the high variance of the sample approximation of the log partition function.
This shortcoming is particularly problematic in the scenario where samples are coming from different batches and some of them have few observations, which is relevant to many biomedical applications.
The proposed approach seems to be well motivated and derived.
While I am not entirely familiar with this literature, the authors seem to have satisfactorily put their contribution in the context of the already abundant literature of extensions of InfoNCE.

**Weaknesses:**

Overall, the experimental part of the paper is a little bit underwhelming for a paper that claims to be rooted deeply into molecular representation for drug screening data, which is a crowded area.
The benchmarks only feature other generic approaches (i.e. CLIP and CCL) and the experimental details are often lacking to ensure a proper reproducibility of the results.
In fact, the initial simulation study has very little to do with drug effect, as the data used there is Gaussian noise (from various, controlled mixtures).
The results of this first part are purely qualitative (plots of low-dimensional representations of the learnt embedding).
As such, I don't find them particularly convincing compared to CLIP or CCL and it would have been nice to have quantitative metrics showing that InfoCORE representations preserve the original structure of the data (e.g. neighbourhoods) within batches but mix them well across batches after integration (see for instance the scANVI paper from Wu et al. 2021).
The second part tackles a specific (somewhat unusual maybe?) task of identifying drugs that are likely to induce a given effect (I find that the details describing this task to be lacking even after going through appendices).
The quantitative results there do not seem to be significantly better than other methods and the experimental setting seems a bit convoluted as it requires adapting the methods benchmarked against in a non-trivial way.
In all tables, most values written in bold do not seem to be significantly better. This is a bit problematic, IMO.
Perhaps more importantly, the paper avoids addressing what exactly is intended by "batch effect" or "confounding factors".
Notably, the treatment of the fact that LINCS data features drug screening of different cell lines is (it is considered a batch effect?) is relegated pretty far into the appendices while I think it is a crucial question.
In general, although a lot of other methods operate similarly, I find that methods that predict drug perturbation without modelling explicitly the state of the cells those perturbations are measured on to be of limited impact (impossible to extrapolate to cells that are not coming from the exact cell lines present in LINCS, let alone on real tissues).
At the end of the day, I think InfoCORE presents interesting novelties and seems to be sound, but I am not entirely convinced it is a great match for drug screening data.
Perhaps the interesting analogy with learning fair representation could be the more important application (where the experimental results also seem to be superior)?

**Questions:**

1) Can the authors comment on Proposition 3? Is this bound tight?
2) At the end of Section 2.2.2, the authors make the interesting comment that "once the batch effect has been mitigated, InfoCORE implictly adjusts and ceases to reweigh the negative samples". Is it something that has been observered in practice? It would be great to show some evidence of it.
3) One important motivation for the derivation of InfoCORE is that it should be more robust to settings where some conditions feature few observed samples. Do you have further experimental results that characterizes how this is the case (against CCL for instance)? For instance, it could be interesting to control the number of samples from one or a few "rare" conditions, or at least to show the performance on test samples from such rare conditions.

---

> ### Author Response · Authors · 2023-11-17
> **Response to Reviewer g7wR**
>
> We appreciate the reviewer's detailed feedback, which helped improve our paper. The reviewer’s focus on our experiments inspired expanded discussions and enhancements in our revised manuscript and appendix. Detailed explanations of these changes are provided below.
>
> **1. About drug representation experiments**
>
> >Experiments are underwhelming and benchmarks are generic approaches.
>
> Although molecular representation, especially unimodal representation, is a well-explored area, multimodal molecular representation with drug screening data, which can be generalized to new molecular structures without drug screens, is a fairly new field. To the best of our knowledge, there are only a few prior works utilizing cell imaging data to learn molecular representations [1,2], and a few utilizing the gene expression [3,4].
>
> [1] directly applied the generic CLIP method to drug structure and cell imaging data, and [2] applied a similar CLOOME criterion for contrastive learning with the additional masked graph modeling loss. [3] used an over-parametrized autoencoder, and [4] used the standard triplet loss on the L1000 drug screening gene expression data. While these datasets have significant batch effects, e.g., batch number can be predicted from gene expression data, none of these works considered the important batch confounding issue.
>
> [1] Molecule-Morphology Contrastive Pretraining for Transferable Molecular Representation. bioRxiv, 2023.
>
> [2] Cross-modal Graph Contrastive Learning with Cellular Images. bioRxiv, 2022.
>
> [3] Causal network models of SARS-CoV-2 expression and aging to identify candidates for drug repurposing. Nature Communications 2021.
>
> [4] Predicting mechanism of action of novel compounds using compound structure and transcriptomic signature coembedding. Bioinformatics 2021.
>
> >Experimental details are lacking.
>
> We updated and significantly expanded Appendix F and Appendix H to better explain the experimental details. Specifically, in Appendix F, we now explain in detail each of the four adaptations we made to apply prior benchmarking models such as CLIP to drug screening data and we showcased the effectiveness of these adaptations as compared to the vanilla CLIP model via an ablation study. In Appendix H, we provided detailed explanations on the pretraining setup and how accuracy is calculated in the molecule-phenotype retrieval task, as well as the significance of accuracy across each retrieval library.
>
> >Details of task in Table 2 are lacking and results are not significantly better.
>
> In Table 2, the molecule-phenotype retrieval task is a common and essential task corresponding to drug repurposing, which is also evaluated in references [1-3]. Here, we calculate retrieving accuracy over two retrieval libraries, "*whole*" and "*batch*"; "*whole*" comprises all molecules in the held-out set, reflecting the model's overall capacity to match drugs with their effects; in contrast, "*batch*" contains only those molecules from the held-out set sharing the same batch identifier as the target drug, gauging the model's discriminatory ability without batch confounder features. For instance, a model solely reliant on batch-related features might perform well in "*whole*" by randomly choosing drugs from the correct batch, but would likely falter in "*batch*".
>
> Therefore, accuracies **over both libraries as a whole** reflect the model’s ability of molecule-phenotype retrieval for drug discovery and drug repurposing. We updated Section 3.2 and Appendix H accordingly to better explain the setting.
>
> It is important to note that InfoCORE is **the only** model that has strong performance **in both libraries**. In comparison, CCL has very poor performance in "*whole*", and CLIP has significantly worse performance in "*batch*" (for example, in GE, InfoCORE outperforms CLIP over “*batch*” with a 15% increase in top 1 accuracy and 11% in top 5 accuracy).
>
> We also added the standard deviation of the retrieving accuracies over 3 random seeds in Table 6 in Appendix I.
>
> [1] Molecule-Morphology Contrastive Pretraining for Transferable Molecular Representation. bioRxiv, 2023.
>
> [2] Cross-modal Graph Contrastive Learning with Cellular Images. bioRxiv, 2022.
>
> [3] CLOOME: contrastive learning unlocks bioimaging databases for queries with chemical structures. bioRxiv, 2022.

---

> ### Author Response · Authors · 2023-11-17
> **Response to Reviewer g7wR (continued)**
>
> > The experimental setting seems convoluted and it requires adapting the benchmark methods in a non-trivial way.
>
> This is correct and is due to the fact that there are to our knowledge no practically effective multimodal contrastive learning frameworks designed for drug screening data.
>
> While it is not the main focus of our paper, we had to make several novel non-trivial and empirically effective modifications to the generic CLIP and CCL models to adapt and enhance them for drug screening applications, including data augmentation ("*aug*"), cell line specific projection head ("*cellproj*"), training batch sampled by cell line ("*train bycell*"), and adapting MoCo to drug screens data ("*MoCo*"). We updated Appendix F to better explain these adaptations. We also performed an ablation study in the GE dataset to demonstrate the effectiveness of these adaptations.
>
> | Retrieval Library       |      | _whole_ |       |       | _batch_ |       |
> |-------------------------|------|---------|-------|-------|---------|-------|
> | Top N Acc (%)           | N=1  | N=5     | N=10  | N=1   | N=5     | N=10  |
> | InfoCORE                | **6.48** | **19.13**   | **27.53** | **14.16** | **33.93**  | **47.15** |
> | CLIP (reported)         | 6.01 | 18.64   | 27.16 | 12.40 | 30.47   | 42.77 |
> | CLIP (w/o *MoCo*)         | 5.71 | 17.45   | 26.14 | 11.57 | 28.82   | 41.32 |
> | CLIP (w/o *aug*)          | 5.25 | 16.00   | 23.99 | 12.25 | 29.74   | 41.88 |
> | CLIP (w/o *cellproj*)     | 5.39 | 17.36   | 25.77 | 11.62 | 29.58   | 41.78 |
> | CLIP (w/o *train bycell*) | 5.02 | 16.45   | 24.90 | 10.74 | 28.16   | 40.66 |
> | vanilla CLIP            | 4.64 | 14.86   | 22.16 | 11.32 | 28.46   | 40.55 |
>
> Note that the results of CLIP and CCL reported in Table 2 and Table 3 use these adaptations. Compared with vanilla CLIP, InfoCORE has even larger improvements.
>
> > Results in Table 3 are not significantly better.
>
> It is known that the property prediction task in Table 3 is hard and even small gains on these benchmark datasets are significant. For example in reference [1], the proposed model's gains over the baselines are more marginal than in our paper, and in [2][3], the proposed model does not show consistent gains in most datasets.
>
> [1] Uni-Mol: A Universal 3D Molecular Representation Learning Framework. ICLR 2023.
>
> [2] Graph contrastive learning with augmentations. NeurIPS 2020.
>
> [3] Let invariant rationale discovery inspire graph contrastive learning. ICML 2022.
>
> **2. About simulation experiments**
> > Simulation study has very little to do with drug effect.
>
> We agree that the Gaussian distribution does not gene expression data or features from a cell imaging screen. The main purpose of the simulation experiment was to offer an intuitive illustration in a 2-dimensional representation space to show how InfoCORE works to remove biases and extract the real effects (the real effect categories are available since this is a simulation study).
>
> > Quantitative metrics should be added.
>
> We thank the reviewer for this suggestion. We added the following quantitative metrics in addition to the qualitative illustrations. Following the metrics used in scANVI (Wu et al. 2021), we calculated the prediction accuracy (from the latent representation to predict either batch identifier or real effect category), KNN purity, and entropy of batch mixing for each model. The prediction accuracy (%) is as follows.
>
> |                |              | CLIP       | CCL        | InfoCORE       |
> |----------------|--------------|------------|------------|----------------|
> | Drug screens representation   | real effect  | 44.2(7.0) | 45.1(4.6) | **73.3(5.0)** |
> |  | batch id     | 8.6(2.6) | 7.8(1.9) | **5.6(3.6)** |
> | Drug structure representation | real effect  | 61.4(11.7) | 48.0(10.2) | **77.8(6.0)** |
> |  | batch id     | 14.4(4.5) | 11.7(3.4) | **5.4(2.4)** |
>
> We updated the results with all three metrics in Figure 3 and Figure 4. We also updated Section 3.1 and Appendix I to better explain the metrics and results. InfoCORE outperforms CLIP and CCL, especially in prediction accuracy of real effect category and KNN purity.

---

> ### Author Response · Authors · 2023-11-17
> **Response to Reviewer g7wR (continued 2)**
>
> **3. About problem setting**
> > The paper avoids addressing what exactly is intended by "batch effect" or "confounding factors".
>
> We added the following sentence in the introduction: “Batch effects refer to non-biological effects introduced into the data through the experimental measurement process.” See e.g. [1] as well as our explanations in Section 2.1. Specifically, in the context of drug screening, each “batch” refers to a (384-well) plate used in the experiments (384 drugs are tested at once in one plate; experiments on different plates are performed e.g. by different people, on different days, etc. and thus show non-biological variation, termed “batch effect”; however, complicating the situation is the fact that variation between plates is not always non-biological as we discussed in Section 2.1; this is due to the fact that similar drugs are often tested on the same plate and different cell lines are tested in different plates).
>
> [1] Adjusting batch effects in microarray expression data using empirical Bayes methods. Biostatistics, 2007.
>
> > The different cell lines screened in LINCS data is a crucial question.
>
> We agree that identifying cell line specific drug effects and their interaction is an important question. However, the main focus of this paper is to learn a generic drug representation that can be generalized to new drug structures without performing additional drug screens. Also, note that screening was only performed in one cell line in the CP dataset, and only 9 cell lines have sufficient data in the GE dataset. This is not sufficient to extrapolate to new cell lines and we therefore do not consider this task.
>
> **4. Question 1**
> >Can the authors comment on Proposition 3? Is this bound tight?
>
> The bound is tight when $\hat{p}(x_b^1|z_g^1)=\hat{p}(x_b^1|z_d^1)$.
>
> **5. Question 2**
> > Empirical evidence of InfoCORE's adaptive adjusting procedure.
>
> We observed evidence of InfoCORE’s adaptive reweighting schedule in practice.
> * We tracked the entropy of the (normalized) reweighting factors during training. A higher entropy indicates more uniform reweighting. For example, in the GE dataset, the entropy of the reweighting factors increases from 9.99 to 10.26 (in comparison, CLIP, i.e. uniform, has entropy 11).
> * We tracked the cross-entropy loss of the batch classifiers. A higher loss implies fewer batch-related features. For example, in the GE dataset, the cross entropy loss of the classifier based on drug structure representation first quickly decreases from 4.8 (random initialization) to 2.7 as we optimize the classifier’s parameters, and then slowly increases to 3.9.
>
> **6. Question 3**
> > Performance under rare conditions.
>
> We would like to clarify that in both drug screening datasets, experiments are done using 384-well plates. Thus there are only about 300 drugs in each batch (i.e., plate, since the remaining wells are used as controls) and hundreds of batches in total. Thus, every batch condition is “rare”. This is the reason for the poor retrieving accuracy of CCL over the drug library “*whole*”.
>
> We thank the reviewer for proposing an additional experiment for further validation. We experimented with the CP dataset and downsampled the data to a third in 5 (out of 97) randomly selected batches that are correlated with other batches. The average retrieving accuracy within these rare batches is shown below. As expected, the performance drop of InfoCORE is about 3%-4% less than that of CCL after downsampling.
>
> | Top N Acc | training dataset | N=1         | N=5         | N=10        |
> |-----------|---------------|-------------|-------------|-------------|
> | CCL       | normal        | 9.45        | 34.55       | 52.00       |
> |           | downsample    | 6.18        | 28.73       | 43.64       |
> |           | % diff        | -34.60%     | -16.85%     | -16.08%     |
> | InfoCORE  | normal        | 10.55       | 33.09       | 48.00       |
> |           | downsample    | 7.27        | 29.09       | 41.82       |
> |           | % diff        | **-31.09%** | **-12.09%** | **-12.88%** |

---

> ### Comment · Area_Chair_wfRY · 2023-11-20
> **Respond to authors' rebuttal**
>
> Please, confirm that you have read the author's response and the other reviewers' comments and indicate if you are willing to revise your rating.

---

> ### Comment · Reviewer_g7wR · 2023-11-21
>
> For clarity, my comment about the significance of the improvement was not meant to be understood as a judgement on how large the estimated improvement is compared to prior art, but as whether they are statistically significant (i.e. is the improvement really due to better performance of the method or due to the randomness of the evaluation / estimation process). In many cases, the estimated improvement is smaller than the standard deviation of the estimator, hence the lack of significance.
> I do agree that it can be very difficult to achieve a significant improvement in such a crowded task, though.
>
> While I agree that 9 cell lines is insufficient to infer a general model that would extrapolate to new cell lines would be unreasonable, this is exactly my concern about the general approach that "learn(s) a generic drug representation that can be generalized to new drug structures". By not taking into account the perturbed cells in the modelling, you are implicitly trying to learn a representation that is universal, or at least agnostic to the cell line the drug is applied on. This is a much harder task that modelling the cell line and limiting the scope to predicting perturbations for those 9 cell lines.
>
> Otherwise, I want to thank the authors for their careful reply and addressing the comments I made in my review. I think that the quality of the manuscript is now significantly better and allows for better reproduction of the results. I will improve my rating of the paper accordingly.

---

> ### Author Response · Authors · 2023-11-21
> **Response to the Latest Comment by Reviewer g7wR**
>
> We appreciate the reviewer's constructive feedback and their increased score. We are glad our revisions have addressed most of their concerns. The reviewer's point about considering perturbed cells for understanding drug mechanisms in different contexts is valid, and we will clarify our model's scope in relation to cell lines in our discussion.

---

### Author Response · Authors · 2023-11-17
**General Response and Summary of Key Revisions**

We thank all the reviewers for their constructive feedback that helped us improve our paper. Based on these comments, we have revised our manuscript, with the changes highlighted in red. Below we provide a brief summary of the key revisions:

* We revised the introduction and method section to further clarify the problem setting and methodology. We added a table summarizing the definitions of all variables used in our paper in Appendix J.
* We added three quantitative metrics for the simulation experiments in Figure 3 and Figure 4, and provided a detailed explanation of the metrics and results in Appendix I, including prediction accuracy, KNN purity, and entropy of batch mixing.
* We provided expanded experimental details and we conducted an ablation study of each of the four adaptations we made to enable application of the vanilla benchmarking models to drug screening data in Appendix F and H.

We also added standard deviations of the retrieving accuracy results based on 3 random seeds in Appendix I, and we updated the results in Table 2 with the average over the 3 runs. Furthermore, we corrected a bug in our code for the CCL baseline, which led to slightly worse results for CCL; see Table 2 and Table 3 of the revised manuscript. Note that all our initial empirical observations still hold true.

---

### Meta-Review · Area_Chair_wfRY · 2023-12-07

**Metareview:**

Summary:

This paper studies multimodal molecular representation learning. The authors analyze the confounding batch effects to the learned molecular representations, and adopt a conditional mutual information maximization objective among the modalities, called the Information maximization approach for COnfounder Removal (InfoCORE). InfoCORE also reweighs the negative samples in the InfoNCE objective differentially based on the posterior batch distribution, to resolve the issue of insufficient negatives. Empirical experiments were conducted to evaluate the proposed method along with baselines on one simulated dataset and two real drug screening datasets for two downstream tasks, including molecule-phenotype retrieval and molecular property prediction. A real-data experiment was presented to demonstrate that InfoCORE can be applied beyond drug discovery, improving fairness metrics when learning representations with
sensitive attributes.

Strengths:

- InfoCORE is addressing an important shortcoming of InfoNCE in the scenario where samples are coming from different batches and some of them have few observations.
- The proposed approach seems to be well motivated and derived.
- The authors seem to have satisfactorily put their contribution in the context of the already abundant literature of extensions of InfoNCE.
- The studied problem is important.
- The proposed method is interesting.
- The paper is well written.
- Review of prior work is given and comparison to prior work is clear, walkthrough of derivation is intuitive and easy to follow.
- Figures of the graphical model and computational workflow are presented in a clean fashion that aids understanding the framework.
- The method is adequately novel and addressed the negative sample supply problem in conditional contrastive learning.
- Thorough experiments.
- The use of reweighting the negative samples differentially based on the posterior batch distribution solves the challenge of poor generalization in the case of limited negative samples for CCL.
- The results show a clear improvement in the metrics for both simulated data and real data tasks.

Weaknesses:

- The benchmarks only feature other generic approaches (i.e. CLIP and CCL) and the experimental details are often lacking to ensure a proper reproducibility of the results.
- The initial simulation study has very little to do with drug effect, as the data used there is Gaussian noise (from various, controlled mixtures).
- The results of this first part are purely qualitative (plots of low-dimensional representations of the learnt embedding) and not particularly convincing compared to CLIP or CCL.
- It would have been nice to have quantitative metrics showing that InfoCORE representations preserve the original structure of the data within batches but mix them well across batches after integration.
- Some quantitative results there do not seem to be significantly better than other methods and the experimental setting seems a bit convoluted.
- The computational considerations should be illustrated.

Recommendation:

A majority of reviewers lean towards acceptance. I, therefore, recommend accepting the paper and encourage the authors to use the feedback provided to improve the paper for the camera ready version.

**Justification For Why Not Higher Score:**

Most reviewers only lean slightly towards acceptance. Paper still has several weaknesses:

- The benchmarks only feature other generic approaches (i.e. CLIP and CCL) and the experimental details are often lacking to ensure a proper reproducibility of the results.
- The initial simulation study has very little to do with drug effect, as the data used there is Gaussian noise (from various, controlled mixtures).
- The results of this first part are purely qualitative (plots of low-dimensional representations of the learnt embedding) and not particularly convincing compared to CLIP or CCL.
- It would have been nice to have quantitative metrics showing that InfoCORE representations preserve the original structure of the data within batches but mix them well across batches after integration.
- Some quantitative results there do not seem to be significantly better than other methods and the experimental setting seems a bit convoluted.
- The computational considerations should be illustrated.

**Justification For Why Not Lower Score:**

Most reviewers lean towards acceptance.

---

### Decision · Program_Chairs · 2024-01-16

Accept (poster)